# Integrated reconstructive spectrometer with programmable photonic circuits

Chunhui Yao[1], Kangning Xu[2], Wanlu Zhang[1], Minjia Chen [1], Qixiang Cheng [1,2] ✉ & Richard Penty[1]

Optical spectroscopic sensors are a powerful tool to reveal light-matter interactions in many fields. Miniaturizing the currently bulky spectrometers has become imperative for the wide range of applications that demand in situ or even in vitro characterization systems, a field that is growing rapidly. In this paper, we propose a novel integrated reconstructive spectrometer with programmable photonic circuits by simply using a few engineered MZI elements. This design effectively creates an exponentially scalable number of uncorrelated sampling channels over an ultra-broad bandwidth without incurring additional hardware costs, enabling ultra-high resolution down to single-digit picometers. Experimentally, we implement an on-chip spectrometer with a 6-stage cascaded MZI structure and demonstrate <10 pm resolution with >200 nm bandwidth using only 729 sampling channels. This achieves a bandwidth-to-resolution ratio of over 20,000, which is, to our best knowledge, about one order of magnitude greater than any reported miniaturized spectrometers to date.

The past decades have witnessed an ever-growing demand for in situ, in vitro and in vivo spectroscopic measurement techniques to facilitate various applications, ranging from wearable healthcare monitoring, portable chemical sensing tools, to compact optical imaging systems[1,2]. This trend has propelled the rapid progression of miniaturized optical spectrometers in both academia and industry. The development of mini- or micro- spectrometers follows the same principles as for their benchtop counterparts[3,4], including wavelength demultiplexing spectrometers (WDSs) with dispersive elements or narrowband filters, and spatial or temporal heterodyne Fourier transform spectrometers (FTSs). With the advances in nano-fabrication technologies, these schemes were later applied to photonic integrated circuits (PICs) for on-chip spectrometry, enabling significantly reduced device footprints[5]. Nevertheless, due to physical constraints on available chip area, number of building blocks, and/or power consumption, their performance, especially the resolution and bandwidth, is inevitably bounded. For example, the WDSs, including those based on arrayed waveguide gratings[6], echelle diffraction gratings[7], or tunable narrowband resonantors[8–10], spectrally decompose the incident light

into spatial or temporal detection channels. This results in a strict one-to-one linear mapping between the spectral pixels and channel number. Consequently, the pixel number, i.e., the bandwidth-to-resolution ratio, is ultimately bounded by the minimum detectable power per channel. On-chip FTSs also suffer the trade-off between performance and resource consumption, as the resolution is inversely proportional to the optical path length or the maximum driving voltage, while the bandwidth depends on large interferometer arrays or complex wavelength-division-multiplexing (WDM) strategies[11–13]. A clear performance boundary can be seen that the reported on-chip WDSs and FTSs to date typically have a moderate bandwidth-to-resolution ratio in the range from tens to hundreds, with a few exceptions reaching a few thousands[14–16].

In recent years, the emergence of reconstructive spectrometers (RSs) have revolutionized this field by leveraging computational algorithms to ease the hardware burden in conventional schemes[17]. Specifically, RSs follow a global sampling strategy whereby they use a limited number of sampling channels encoded with varied spectral responses to sample the entire incident spectrum and recover a larger

[1]Centre for Photonic Systems, Electrical Engineering Division, Department of Engineering, University of Cambridge, Cambridge CB3 0FA, UK. [2]GlitterinTech Limited, Xuzhou 221000, China. ✉e-mail: qc223@cam.ac.uk

number of spectra pixels by solving inverse problems. This naturally facilitates the development of on-chip RSs as fewer resources are required. For example, various RS designs have been proposed based on nanophotonic structures, such as random scattering media[18–20], metasurfaces[21], or photonic crystal-based filter arrays[22], to allow compact footprints. However, as they require the incident light to be passively split into many physical detection channels, the channel numbers are inevitably restricted due to the power splitting loss. Researchers have also explored active RSs using lumped structures with tunable sampling spectral responses, including detector-only RSs with tunable absorption spectra[23,24], filter-based RSs with MEMS[25] and thermally tunable resonance cavities[26,27]. However, as the sampling channels are generated by setting different levels of driving powers to alter the waveform, they inevitably suffer from a poor decorrelation with each other, resulting in a compromised resolution (typically on the order of nanometers[27,28]) and limited scalability. In our earlier work, we introduce an RS design that utilizes a reconfigurable network with distributed micro-ring resonator filters to create well-decorrelated sampling responses, while its bandwidth is constrained by the switching network, and the resolution has a trade-off against device complexity[29]. So far, the reported on-chip RS schemes haven't shown a definitive performance advantage over the WDSs and FTSs.

In this paper, we first reveal that in order to fully harness the benefits of a global sampling strategy in RSs and facilitate ultra-high spectroscopic performance, a substantial number of broadband sampling channels with desirable spectral responses is necessary. To fulfill this target while minimizing the on-chip resource consumption, we present a powerful RS scheme based on a programmable photonic sampling circuit comprising multiple stages of tunable interferometers. In our design, each interferometer's spectral properties are engineered to achieve an overlaid transmission spectrum with rapid pseudo-random fluctuations, enabling a small auto-correlation width that enhances global sampling efficiency. By programming the phase shift of each interferometer individually, the resultant transmission spectra can be temporally decorrelated, enabling sampling channels with low cross-correlation. More attractively, in such a cascading configuration, the total number of sampling channels (corresponds to the overall combination of phase shifts), can exponentially expand with an increase in either the number of interferometer stages or phase states per interferometer, while improving the sampling performance per channel. Thereby, with only a few waveguide components, hundreds or even thousands of high-performance sampling channels can be efficiently generated to escalate the spectrometer resolution, reaching down to single-digit picometers. Experimentally, we implement a 6-stage device with unbalanced Mach-Zehnder interferometers (MZIs) on a commercial Silicon Nitride (SiN) photonic integration platform and show that 729 sampling channels are sufficient for attaining an ultra-high resolution of <10 pm over a broad bandwidth of >200 nm, i.e., a bandwidth-to-resolution ratio of over 20,000. This is, to our best knowledge, about one order of magnitude greater than any reported miniaturized spectrometers, being comparable to or even exceeding many commercial benchtop spectrum analyzers[30]. It is, however, noted that the experimentally resolved resolution and bandwidth are limited by the available lab equipment. We thus further investigate the device's capability via rigorous simulations that predict <5 pm resolution. Meanwhile, by utilizing dispersion-engineered waveguide components, we showcase that the device bandwidth could be readily enhanced to >400 nm. The reconfigurability of our design offers users the versatility to customize the device to achieve performance trade-offs on resolution, reconstruction accuracy, sampling time, and computational complexity by grouping different combinations of sampling channels, covering the application scenarios from identifying signature spectral peaks with acceptable levels of performance[31] to relative metrology with ultra-high resolution and low errors[32].

## Results
### Principle and design

For RSs, the output power intensity $I$ of an unknown incident spectrum $\Phi(\lambda)$ propagating through a broadband sampling channel with a transmission spectral response $T(\lambda)$ can be described as:

$$I = \int T(\lambda)\Phi(\lambda)d\lambda \tag{1}$$

Likewise, $M$ sampling channels encoded with distinctive spectral responses correspond to $M$ power intensities to be detected, such that the Eq. (1) can be discretized and expressed in a matrix format, as:

$$I_{M\times 1} = T_{M\times N}\Phi_{N\times 1} \tag{2}$$

where $N$ denotes the number of spectral pixels in the wavelength domain, while $T_{M\times N}$ is the well-known transmission matrix. Accordingly, $\Phi(\lambda)$ can be reconstructed by solving the inverse problem defined by Eq. (2), even under the case when the sampling number $M$ is considerably smaller than the dimension of incident spectrum $N$ (i.e., an underdetermined problem where $M\ll N$), which points out the advantage of the global sampling strategy of RSs. Conventional ways of solving an inverse problem can be expressed as[33]:

$$Minimize\ |I - T\Phi|_2\ subject\ to\ 0 \le \Phi \le 1 \tag{3}$$

In practice, to reduce the ill-conditioning of the undetermined problem, Eq. (3) is often modified by adding additional regularization terms, such as the modified Tikhonov regularization[34], to be written as:

$$Minimize\ |I - T\Phi|_2 + \alpha\ |\Gamma\Phi|_2\ subject\ to\ 0 \le \Phi \le 1 \tag{4}$$

where $\alpha$ is the regularization weight and $\Gamma$ is a difference-operator that calculates the derivative of $\Phi$, which helps effectively mitigate the reconstruction errors induced by noise. The concept of compressive sensing (CS) can be borrowed to help understand the reconstruction in underdetermined systems (please find more discussions in Supplementary Section S1). As revealed by CS theories, the minimum number of sampling channels (i.e., $M$) required to reconstruct an N-dimension incident spectrum is proportional to $A \log N$, where $A$ is a constant related to the design of transmission matrix[35,36]. In other words, a properly engineered transmission matrix could leverage a large number of spectral pixels with limited sampling channels, hence enabling a high bandwidth-to-resolution ratio. Numerically, we use auto-correlation and cross-correlation to assess the performance of transmission matrices. The auto-correlation function of a transmission matrix is written as[18]:

$$C(\triangle\lambda) = \left\langle \frac{\langle I(\lambda,k)I(\lambda+\triangle\lambda,k)\rangle_\lambda}{\langle I(\lambda,k)\rangle_\lambda\langle I(\lambda+\triangle\lambda,k)\rangle_\lambda} - 1 \right\rangle_k \tag{5}$$

where $I(\lambda,k)$ is the transmission intensity at channel $k$ for wavelength $\lambda$, and $\langle\cdots\rangle$ corresponds to the average over wavelengths or channels. The half-width-half-maximum (HWHM) of $C(\triangle\lambda)$, namely the auto-correlation width $\delta\lambda$, represents the wavelength shift that is sufficient to reduce the degree of correlation of individual sampling channels by half, indicating the capability of distinguishing adjacent wavelength pixels. Thus, $\delta\lambda$ is usually regarded as an important performance indicator for the sampling channels[37]. However, it should be noted that the resolution of the spectrometer system is jointly determined by many other factors including number of sampling channels, algorithm performance, signal-to-noise ratio (SNR), and etc.[29,38,39]. In the following, we show that the resolved resolution can significantly exceed the value of $\delta\lambda$ by employing a sufficient number of sampling channels.

Cross-correlation, on the other hand, denotes the similarity between different channels.

Mathematically, the most ideal transmission matrix should follow an independent random distribution[36,37], i.e., the transmission intensities at each specific wavelength are discretely independent among channels to ensure that both the value of δλ and the averaged cross-correlation can be suppressed close to zero. This, however, cannot be physically achieved since the spectral response of any photonic structures is continuous. Therefore, to explore the performance boundary of RSs as a guide for practical designs, we randomly generate a group of broadband, continuous sampling channels with rapid spectral fluctuations over a 200 nm wavelength range (hence featuring low δλ and cross-correlation) to approximate the ideal transmission matrix, as shown in Supplementary Fig. 1a. The δλ of all matrices is set at 0.2 nm, while the averaged cross-correlation is below 0.05, serving as a reasonable benchmark for reconstructive spectrometers[18,20,26]. Detailed procedures for generating these matrices are provided in Supplementary Section S1. Accordingly, we first test the relationship between the number of sampling channels and the RS resolution by solving dual spectral lines located at two adjacent wavelength pixels (the Rayleigh Criterion, i.e., the resolution being equal to $\lambda_{bandwidth}/N$) based on Eq. (4). Here, CVX convex optimization algorithm is used for the spectrum recovery in simulations[40]. Figure 1a plots the number of sampling channels required to reconstruct varying numbers of spectral pixels. It can be seen that as the target spectral pixel $N$ increases, i.e., the resolution gradually improves, the required channel number $M$ ascends in a stepwise fashion. This finding agrees with the CS theory as $M \sim \mathcal{O}(A \log N)$. Specifically, when $N$ is relatively small and $A$ could be considered as a low-value constant, $M$ exhibits a gradual but slow growth as $N$ expands over long intervals, reflecting the logarithmic relationship between $M$ and $N$. As $N$ grows larger, the changes in $A$ become more pronounced, such that even a small increment in $N$ results in a substantial increase in $M$ (see further discussions in Supplementary Section S1). For example, as illustrated in Fig. 1a, a 12 pm resolution can be achieved using around 50 sampling channels; however, to further improve the resolution to <9 pm, more than 300 channels are needed. Besides, we also investigate the impact of channel number on the reconstruction error by recovering a broadband input spectrum (with a constant spectral pixel number $N$ set as 5000). Simulation details are elaborated in Supplementary Section S1. Here, the L2-norm relative error $\varepsilon$ is utilized to quantify the spectrum reconstruction accuracy as a widely adopted metric[2,10], which is defined as follows:

$$\varepsilon = \frac{||\Phi_0 - \Phi||_2}{||\Phi_0||_2} \tag{6}$$

where $\Phi$ is the reconstructed spectrum and $\Phi_0$ is the reference. Notably, a relative error lower than 0.1 is typically considered as a strict benchmark indicative of high accuracy[10,26,29,38,41]. As shown by Fig. 1b, the relative error $\varepsilon$ continuously decreases with an increasing number of sampling channels, reaching <0.02 when the channel number exceeds 200.

The above simulation results clearly illustrate the necessity of scaling to hundreds, or even thousands of high performance sampling channels to approach an RS performance with ultra-high resolution and accuracy. To efficiently meet such a requirement, here, we propose a programmable design cascading multiple stage of tunable interferometers. Figure 2a schematically shows the implementation of our design via a series of 1 × 1 unbalanced MZIs. In such a system, the transmission spectrum of an arbitrary MZI stage $i$ can be described as[42]:

$$T_i = \rho^2 + (1-\rho)^2 - 2\rho(1-\rho)\cos(\beta\triangle L_i + \delta_i) \tag{7}$$

where the amplitude of the periodic oscillation term $\cos(\beta\triangle L_i + \delta_i)$ is determined by the power spitting ratio $\rho$ of the directional couplers, while its phase is decided by the length difference $\triangle L_i$ between the two arms as well as the relative phase shift $\delta_i$. Therefore, by engineering the spectral properties of each MZI, such as the free spectral range (FSR) and extinction ratio (ER), and phase-tuning of individual MZI stage, the waveform of the overlaid transmission spectrum can be fully manipulated. Specifically, we introduce distinct length differences in various MZI stages to create interferograms with different FSRs, such that the cascaded interferogram no longer displays the original periodicities and exhibits a pseudo-random fluctuation in the wavelength domain, as shown by the insets in Fig. 2a. The ERs are also tailored to maximize the fluctuation range of the cascaded interferogram, facilitating a high sampling efficiency with low excess loss. Thermo-optic (TO) phase shifters are implemented on the MZI arms to tune the phase of each interferogram, thereby de-correlating different sampling channels. Thus, by temporally programming different combinations of phase shifts, the cascaded interferogram can exhibit unique spectra responses, yielding a transmission matrix with low cross-correlation, as shown in Fig. 2b. Further details regarding the MZIs' parameter design, TO phase tuning, and power efficiency can be found in Supplementary Section S2. As the phase shift of each MZI stage can be tuned individually from 0 to 2π, the overall tuning state (i.e., the total number of temporal sampling channels) features an exponential scalability, as:

$$N_{ch} = P_{state}{}^{N_{stage}} \tag{8}$$

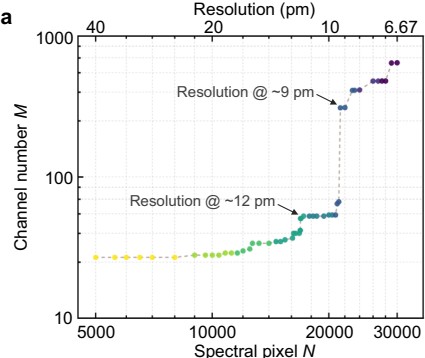

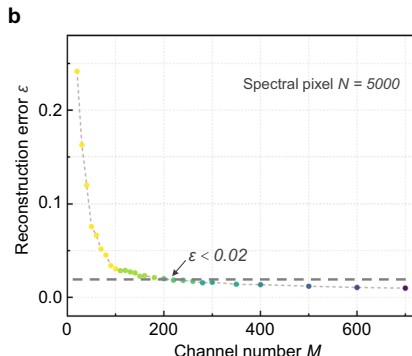

**Fig. 1 | Impact of the sampling channel number on RS performance.** Simulation analyses regarding (**a**) the number of sampling channels utilized to reconstruct varying numbers of spectral pixels (i.e., to achieve different levels of resolution), and (**b**) the relationship between reconstruction errors and the sampling channel number, respectively, based on randomly generated high-performance transmission matrices.

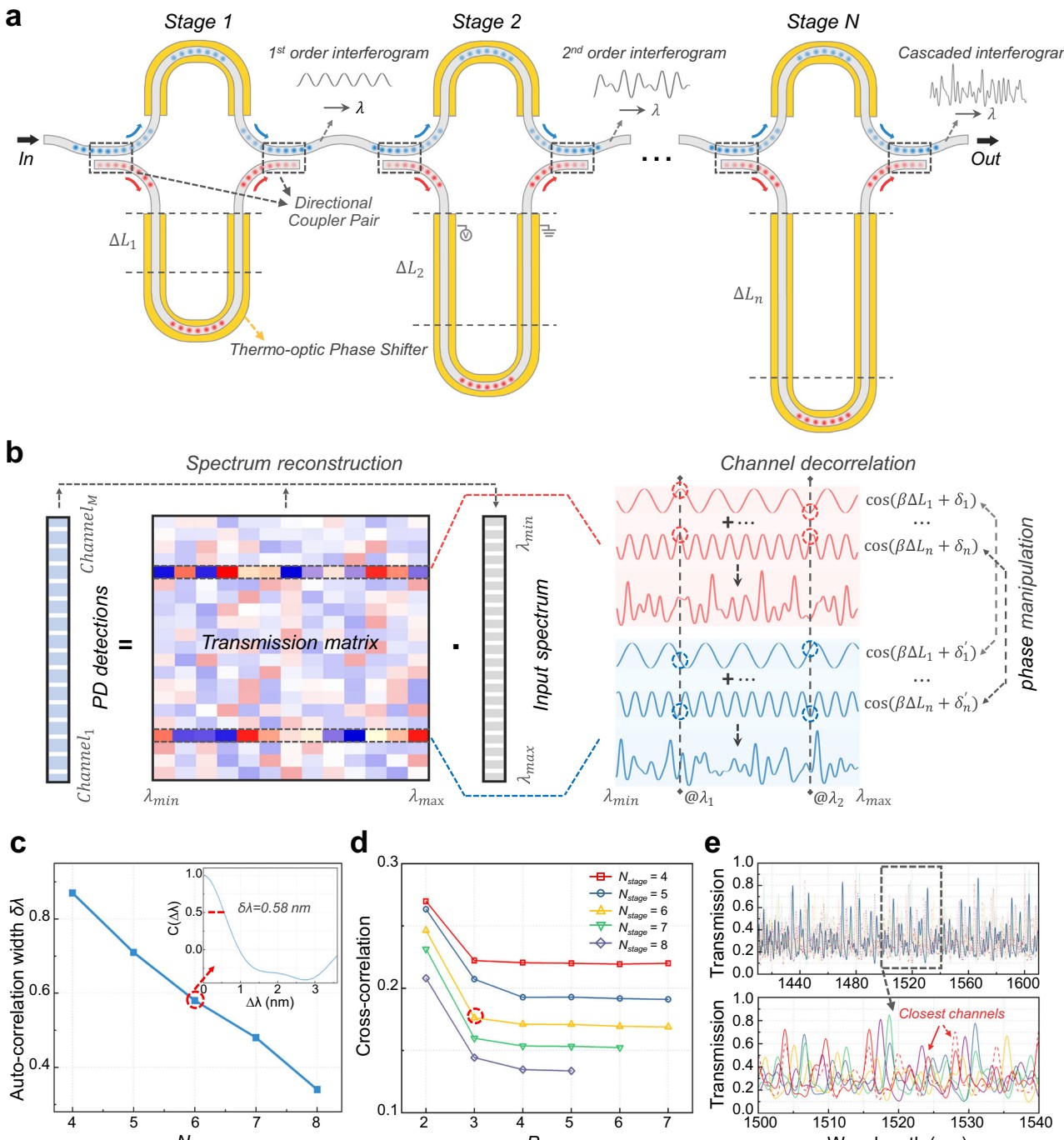

**Fig. 2 | Spectrometer design and simulation. a** Schematic of our proposed programmable spectrometer with multiple stages of unbalanced MZIs, each with a singular input and output port. The insets show the increasing spectral randomness in the cascaded interferogram. **b** Concept illustration of the transmission matrix generated through temporal phase manipulation. The insets depict two exemplary sampling channels with different combinations of phase shifts, thus exhibiting decorrelated spectral responses. The two dashed lines at $\lambda_1$ and $\lambda_2$ highlight the differences of phase shifts. **c** Auto-correlation width δλ of the sampling channels simulated using the spectrometer designs with different stage numbers. The inset shows the auto-correlation function of a 6-stage design with a δλ of 0.58 nm. **d** Averaged cross-correlation between the sampling channels generated by spectrometer designs with varying numbers of MZI stages and phase states per stage. **e** Several example channel spectral responses from a 6-stage spectrometer design with 3 phase tuning states per stage. The red solid and dashed lines represent the closest pair of sampling channels that have diverse phase tuning at only one stage, but still feature a significant spectral difference.

where $N_{ch}$ and $N_{stage}$ represent the channel and stage number, respectively. $P_{state}$ denotes the number of phase tuning states per MZI stage. For example, 4096 sampling channels are easily generated in a 6-stage design with 4 phase states per stage (i.e., 0, π/2, π, and 3π/2).

To further quantify the performance, we simulate the channel spectral responses of our programmable spectrometers with different $N_{stage}$ and $P_{state}$ and calculate their auto- and cross-correlations (see Supplementary Section S2 for more details). As presented by Fig. 2c, the auto-correlation width δλ continuously decreases with a larger stage number, indicating an enhanced sampling efficiency per channel (note that the $P_{state}$ is irrelevant to auto-correlation as phase shifting does not affect channel's waveform). Figure 2d plots the averaged

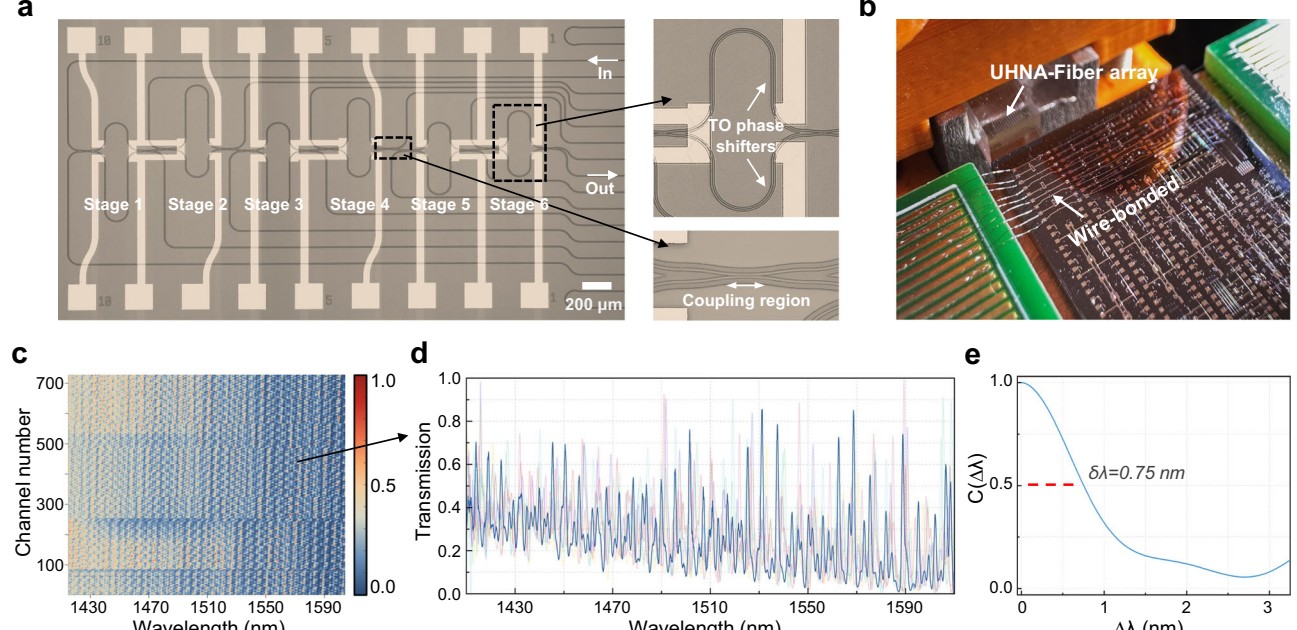

**Fig. 3 | Fabricated spectrometer and its calibration. a** Microscope image of the fabricated on-chip spectrometer in a 6-stage design with enlarged views of a MZI stage and a directional coupler. **b** Photograph of the photonic chip wire bonded to a customized PCB board. An UHNA-fiber array is used for the optical coupling. **c** Normalized transmission matrix of the programmable spectrometer with 729 channels. **d** Transmission spectra of several representative sampling channels, exhibiting the pseudo-random fluctuations in the wavelength domain. **e** The calculated spectral auto-correlation function between 729 channels with a δλ of 0.75 nm.

cross-correlation between one specific sampling channel to all the other channels generated from the designs with varying $N_{stage}$ and $P_{state}$. It can be seen that the averaged cross-correlation also significantly decreases with an increasing number of MZI stage, reaching <0.2 with over 6 stages. A larger $P_{state}$ could also help decrease the cross-correlation, and it gradually converges when $P_{state}$ exceeds 4. Thanks to these attractive features, the sampling channels of our spectrometer could be exponentially scaled up, while enjoying even better sampling performance per channel. Figure 2e plots the simulated transmission spectra of several representative sampling channels in a 6-stage design with 3 tuning states per stage (owning a δλ of 0.58 nm and averaged cross-correlation of 0.176), illustrating a high degree of spectral randomness and sufficient decorrelation between sampling channels.

## Experimental characterization

Figure 3a shows the microscope photo of the fabricated spectrometer. In this demonstration, we implement a 6-stage design with 3 phase settings per stage (i.e., 729 sampling channels in total) as a balanced choice among the device performance, measurement time and computational complexity. The insets present the enlarged views of an MZI stage and a directional coupler. The chip is connected to customized PCB boards via wire-bonding for electrical fan-out, while an Ultra-high-NA (UHNA) fiber array is used for edge coupling, as shown in Fig. 3b. Firstly, we calibrate all the MZIs and derive a look-up table that details the bias voltages required for the phase manipulation of each stage. The transmission matrix is then calibrated by launching the amplified spontaneous emission (ASE) of a semiconductor optical amplifier (SOA) through the chip and measuring the spectral responses for different sampling channels using a commercial spectrum analyzer, as shown in Fig. 3c. Figure 3d plots several examples of the sampling channels, showing the pseudo-random fluctuations in the wavelength domain. Here, the calibration is performed within a 200 nm spectral window from 1410 nm to 1610 nm, which is limited by the bandwidth of the ASE source—the actual bandwidth of the spectrometer is expected

to be larger. Accordingly, all measured spectra are reconstructed over such a 200 nm spectral range. The minor overall downward trend in the transmission can be attributed to the dispersion effect of the directional coupler, which exhibits a slight increase in splitting ratio at longer wavelengths. This can be further improved by employing dispersion-engineered components, such as curved directional couplers, to facilitate a flat splitting ratio over a broader bandwidth range. Detailed discussion of various directional coupler designs can be found in Supplementary Section S2. Moreover, in Supplementary Section S3, we show that by employing curved directional couplers, the device bandwidth could be further extended to over 400 nm. Note that the dispersion from the connecting waveguides has a negligible impact on the device performance, as it is incorporated as part of spectral responses. The δλ of auto-correlation function is calculated to be 0.75 nm, as shown by Fig. 3e, while the averaged cross-correlation is as low as 0.169, both values being in good agreement with the simulation results.

To test the spectrometer performance, we first launch a series of distributed feedback laser diode signals at different wavelengths as narrowband inputs. An electrical control system based on a microcontroller unit is programmed to automatically deploy the bias voltages for the configuration of all sampling channels and collect the real-time output signal intensity from a photodiode. The time for the whole set of measurements is within only 0.8 seconds (see more details about the experimental set-up in Supplementary Section S4). Figure 4a depicts the reconstructed laser signals with the calculated relative error ε ranging between 0.026 to 0.062, indicating high reconstruction accuracy. The full-width-half-maximums (FWHMs) of the resolved peaks at different wavelength locations maintain consistently at about 10 pm. To verify the resolution of our device, we also simultaneously launch two close laser signals as dual-peak inputs. We gradually reduce the spacing between the two spectral lines from 100 pm to 10 pm to conduct a convergence test, as shown by Fig. 4b. It can be seen that all the peak intensities can be well distinguished with a relative error lower than 0.056 at a spacing of 10 pm, illustrating an

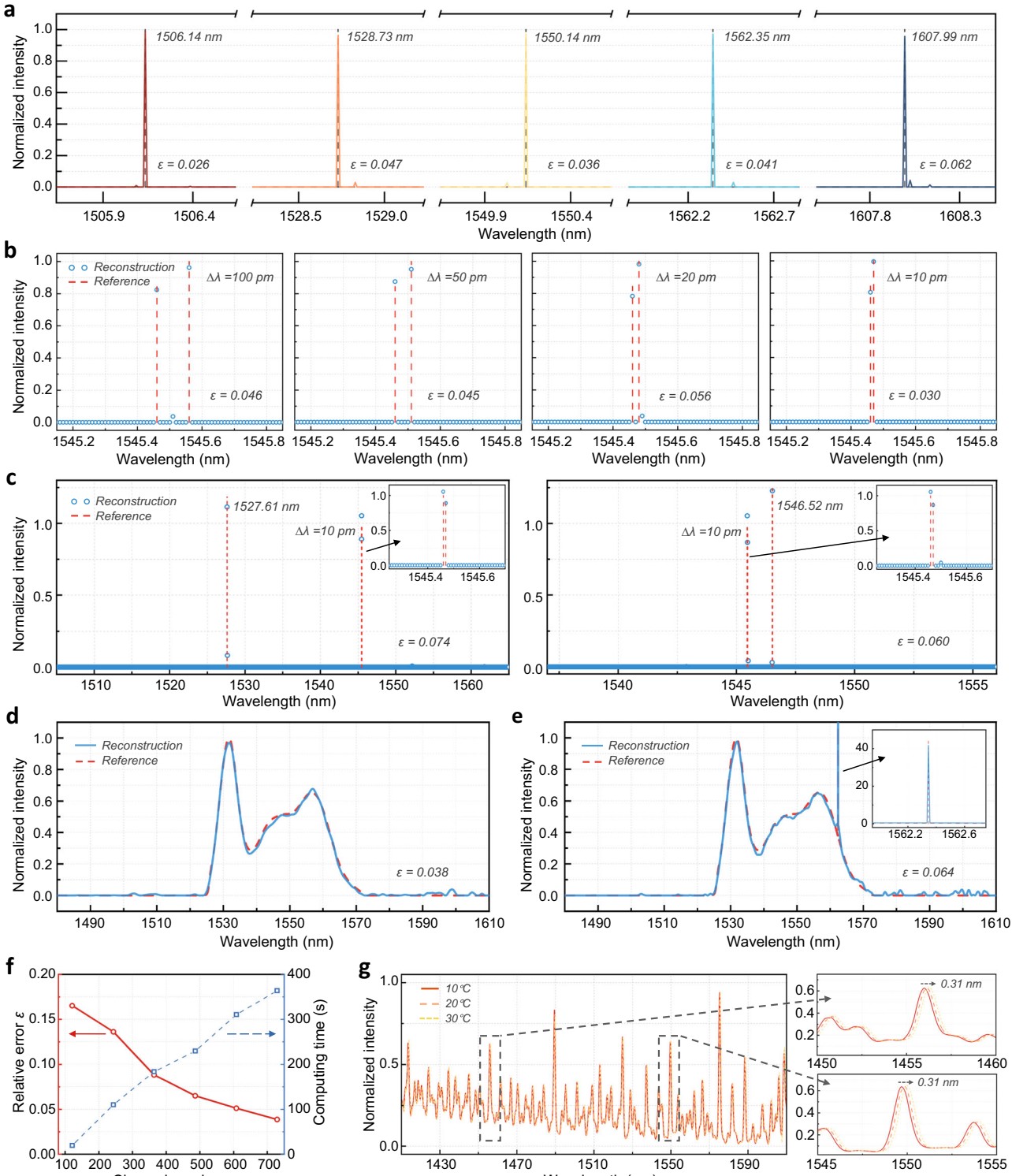

**Fig. 4 | Experimental testing results of the spectrometer. a** Reconstructed spectra for a series of narrowband spectral lines. The black dashed lines mark their center wavelengths. **b** Reconstructed spectra for dual spectral lines with a decreasing spectral spacing from 100 pm down to 10 pm, respectively. **c** Reconstructed spectra for three spectral lines at different wavelengths, with a spectral spacing of 10 pm between two of them. **d** Reconstructed spectrum for a continuous, broadband signal. **e** Reconstructed spectrum for a hybrid signal with both broadband and narrowband spectral components. The inset exhibit the reconstruction of the narrowband component. **f** Investigation regarding the reconstruction accuracy and computing time versus the number of sampling channels. **g** Measured transmission spectra of one specific sampling channel at different temperatures. The insets present the small spectrum redshift as the temperature rises.

ultra-high resolution of that value. Note that we stop such convergence test at 10 pm only due to the limitation of the lab equipment, as no stable spectral spacing at the single-digit picometer level can be generated. However, rigorous simulations on the reconstruction of dual-peak inputs using the same 729 sampling channels indicates a resolution of 8 pm (see Supplementary Fig. 4a). In addition, we show that this resolution could readily achieve <5 pm by programming a larger number of sampling channels, as detailed in the Discussion and Supplementary Section S3. Beyond the dual-peak experiment, we conduct a more challenging triple-peak testing, wherein the spectral spacing among two peaks is set at 10 picometers. Figure 4c shows the resolved three laser peaks with a relative error of 0.074 and 0.060, respectively.

In addition, the reconstruction of a continuous, broadband spectrum is demonstrated by using the ASE spectrum of an EDFA followed by a bandpass filter as input signal. As shown by Fig. 4d, the spectral features are well recovered with a low relative error ε of 0.038. Furthermore, we examine a more challenging case of hybrid spectrum, where both broadband and narrowband signals are presented. To synthesize such incidence, we combine the ASE source and a laser source via a 3 dB fiber coupler. Accordingly, the Eq. (4) is modified by introducing segmented regularization terms, as:

$$\text{Minimize } |I - T(\Phi_1 + \Phi_2)|_2 + \alpha\,|\Phi_1|_1 + \beta\,|\Gamma_2\Phi_2|_2 \text{ subject to } 0 \le \Phi_1, \Phi_2 \le 1 \quad (9)$$

where $\Phi_1$ and $\Phi_2$ denote the narrowband and broadband spectral components, respectively. α and β are the corresponding regularization weights that can be optimized via cross-validation analysis[25]. Figure 4e presents the resolved hybrid spectra with a ε of 0.064, showing that a high reconstruction accuracy can be still achieved.

As previously discussed, the global sampling strategy of RS allows a fewer number of sampling channels to coarsely reconstruct the whole input spectrum. This feature naturally offers our device versatility that trade-offs could be made between various performance metrics, such as the reconstruction accuracy and computational complexity. To investigate their underlying relationship, we analyze the reconstruction error and computing time as a function to the channel number, using the ASE spectrum of EDFA as an example case. Note that here the experiments are conducted by varying the number of phase states across different MZI stages to attain different number of channels. As illustrated in Fig. 4f, the reconstruction error gradually decreases with a larger number of sampling channels, albeit with increasing computing time. Hence, the performance of our design could be user-defined to suit the requirements under different scenarios[17].

For high-resolution spectrometers, the tolerance to temperature variations is also of great concern, as it may severely distort the channel spectral responses[26]. Hence, we implement our design on the SiN platform to take advantage of its low thermal sensitivity relative to other materials such as Si or SiO$_2$. As shown by Fig. 4g, the channel spectral responses of our device redshifts only 0.31 nm for a temperature increase from 10 °C to 30 °C, while the waveform shape remains consistent. Accordingly, we model the thermal stability of our spectrometer as described in Supplementary Section S5. The results reveal that the input spectra can still accurately be reconstructed with a temperature variation up to ± 0.9 °C. In practice, on-chip temperature stabilization techniques can be used to minimize the temperature drift, while the shift in spectral responses could also be offset from an algorithmic perspective through real-time temperature monitoring.

## Discussion

In this paper, we showcase a novel on-chip RS design that facilitates unprecedented spectroscopic performance, using a programmable photonic sampling circuit with cascaded tunable interferometers. The spectral properties of each interferometer are engineered to realize an overlaid transmission spectrum with rapid pseudo-random fluctuations, enabling a small auto-correlation width δλ. By individually programming the phase shift for each interferometer, the system spectral responses can thereby be decorrelated temporally, facilitating sampling channels with low cross-correlation. Such a cascading scheme allows the total number of sampling channel to scale exponentially with the increase in either stage number or phase tuning states per interferometer. Therefore, a massive number of high-performance sampling channels can be readily created on a single chip. Experimentally, we realize an ultra-high resolution of <10 pm over a broad bandwidth of >200 nm, using a 6-stage unbalanced MZI design with only 729 temporal sampling channels programmed. Its performance can be further enhanced by programming a larger number of channels and employing dispersion-engineered waveguide components. We show that a 6-stage design but with curved directional coupler designs can operate over a 400 nm bandwidth with a reconstructed resolution at 5 pm (see Supplementary Section S3 for more details). Leveraging the global sampling scheme, our design features versatility that trade-offs can be made among the reconstruction performance, sampling time, and computational complexity by grouping different combinations of sampling channels, resulting in a user-definable performance to suit different applications.

Demonstrated on a commercial SiN platform, our device is fully compatible with the developed Complementary Metal-Oxide-Semiconductor (CMOS) processes, allowing mass-production at low cost[43,44]. The ultra-high resolution of the proposed device opens doors to numerous applications with critical demands on precision, including the detection of methanol, water pollutants, or gas concentrations, which may require a resolution of <0.1 nm or even 1 pm[2,45–47], and the identification of subtle peak shifts for fiber Bragg grating sensors in monitoring temperature, strain, or chemical variations[48–50].

To highlight the breakthrough accomplished by this work, we present a comprehensive comparison against the state-of-the-art competitors. Figure 5a plots the resolution and bandwidth (as two most important metrics) of integrated spectrometers based on dispersive optics, narrowband filtering, Fourier transform and computational reconstruction, respectively, along with our device[7,9,11,13,14,16,18,21,28,29,39,51–63]. Note that only reported spectrometers with claimed resolutions that are experimentally verified by dual-peak testing (i.e., the Rayley criterion) with high reconstruction accuracies are included. As can be seen, the previously reported spectrometers exhibit clear trade-offs between the bandwidth and resolution, resulting in bandwidth-to-resolution ratios that mostly range from tens to hundreds. Recent demonstrations based on narrowband filtering have made improvements on this to a few thousands but require long sampling times and heavily rely on the electrical control. In comparison, the proposed programmable spectrometer decouples such a trade-off and achieves a record-high bandwidth-to-resolution ratio of 20,000, which can be improved further. We also evaluate the spectrometer performance in terms of the ratio between the number of spectral pixel (denoted as $N_{spectral}$) to device footprint, and the ratio of the number of spectral pixels to the number of spatial detection channel (denoted as $N_{detection\ channel}$), as shown by Fig. 5b. The two terms offer valuable insights showing the capacity of spectral pixels that can be accommodated per unit area or per individual detector, respectively[16]. Our device exhibits a record-high spectral pixel-to-spatial channel ratio, thanks to its excellent reconstruction capability and the utilization of only one physical detection channel. As for the spectral pixel-to-footprint ratio, our device also exceeds the performance of most of the competitors. It should be noted that we specifically select the SiN platform as it offers a superior tolerance to the temperature variations (±0.9 °C), especially when comparing to other spectrometer designs with ultra-high resolutions (see Table S1 in Supplementary Section S6 for detailed comparison). However, this trades off the device footprint (about 1.9 × 3.7 mm$^2$) since the SiN platform has a smaller index contrast comparing to the Si platform. By

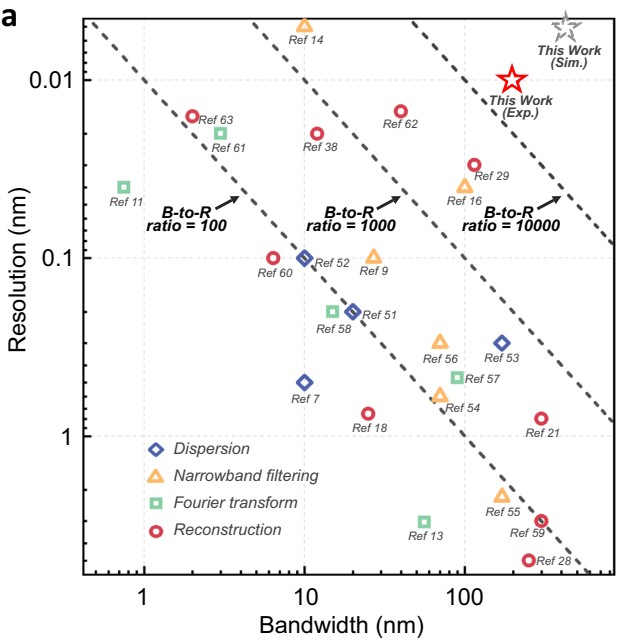

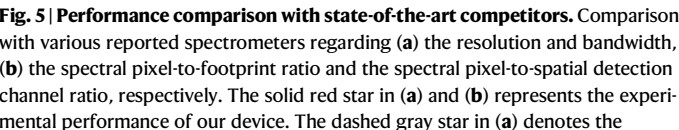

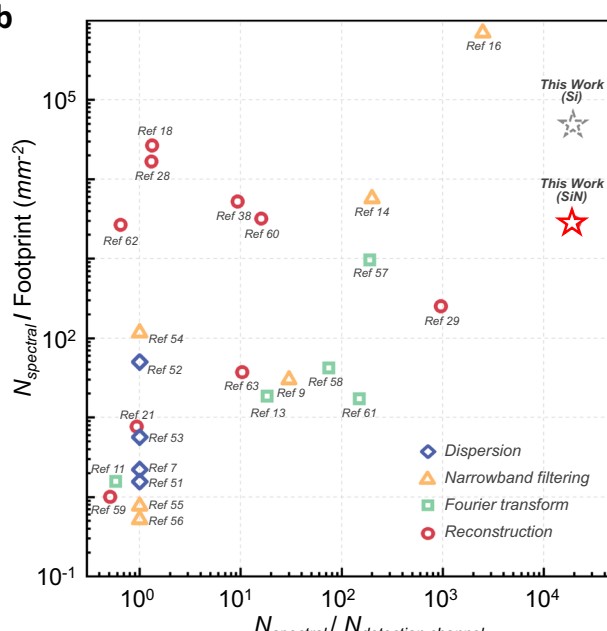

**Fig. 5 | Performance comparison with state-of-the-art competitors.** Comparison with various reported spectrometers regarding (**a**) the resolution and bandwidth, (**b**) the spectral pixel-to-footprint ratio and the spectral pixel-to-spatial detection channel ratio, respectively. The solid red star in (**a**) and (**b**) represents the experimental performance of our device. The dashed gray star in (**a**) denotes the simulation performance when applying a larger number of sampling channels and curved directional couplers, while the dashed lines display the bandwidth-to-resolution ratios at different orders of magnitude. The dashed gray star in (**b**) represents the expected performance when applying the same spectrometer design to a standard 220 nm Si platform.

implementing the same design on a standard 220 nm silicon-on-insulator (SOI) platform[43], the footprint is anticipated to be reduced to <0.4 mm², which would further enhance the spectral pixel-to-footprint ratio by an order of magnitude, as illustrated in Fig. 5b.

In summary, we believe the proposed design offers a promising pathway for the future development of low-cost, mass-manufacturable, chip-scale spectroscopic devices. Its ultra-high performance makes it hold great promise to play a role in a vast range of application areas.

## Methods

### Device fabrication, design, and loss analysis
The spectrometer was fabricated via a CORNERSTONE SiN multi-project wafer (with a 300 nm LPCVD SiN layer) run using standard DUV lithography (250 nm feature size). The waveguides are designed to support TE polarization with a width of 1200 nm. The edge couplers are designed to have a mode diameter about 3.5 μm to match the UHNA fiber array, resulting in a coupling loss of <3.2 dB per facet. The measured on-chip loss of the 6-stage device is about 4.9 dB, corresponding to a loss of roughly 0.8 dB per stage.

### Numerical calculations and simulations
The spectrum reconstructions are performed by running the CVX optimization algorithm on MATLAB, based on a Xeon 7980 CPU with 64 GB memory. ANSYS Lumerical FDTD is used to perform the physical simulations of the directional couplers and edge couplers, while the system spectral responses of the programmable spectrometers are simulated in the ANSYS Lumerical INTERCONNECT module.

### Electrical and thermal control
A microcontroller unit (MCU) is programmed to automatically generate the control signals based on the pre-calibrated voltage look-up table, which is then sent to a high-resolution multi-channel digital-to-analog converter (DAC). The DAC thereby produces the analog electoral signals that are subsequentially amplified by a

customized driving board and injected into the spectrometer chip, enabling the temporal phase manipulation. The real-time output signals from the photodiode are collected using an analog-to-digital converter (ADC) module embedded in the MCU. To stabilize the global temperature during our measurements, a thermoelectric cooler is positioned beneath the chip with the temperature fixed at 25 °C.

## Data availability
All data pertaining to the device design can be found in the article and its Supplementary Information. The data that support the plots within this article are available in the University of Cambridge Repository at https://doi.org/10.17863/CAM.101562.

## Code availability
The code used in this study is available from the corresponding author upon request.

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

## Acknowledgements

This work was supported by UK EPSRC, project QUDOS, under Grant EP/T028475/1. The authors thank CORNERSTONE - University of Southampton for providing free access to their SiN MPW run, funded by the CORNERSTONE 2 project under Grant EP/T019697/1. The authors also thank Mr. Peng Bao for the help in experiments. C.Y. acknowledges the financial support provided by the CSC-Trust Scholarship for his doctoral studies.

## Author contributions

C.Y. conceived the spectrometer design, performed the optical simulations, drawn the chip layout, carried out the experimental measurements with W.Z.'s assistance, and analyzed the experimental data. K.X. conducted the numerical simulations regarding the impact of sampling channel number on the reconstruction performance. W.Z. and M.C. developed the automatic electrical control system. C.Y. and Q.C. wrote the manuscript, with K.X. and R.P. inputs. Q.C. and R.P. supervised the project.

## Competing interests

The GlitterinTech Limited declares a filed pending patent application with the China National Intellectual Property Administration (inventors: C.Y. and T.Y., application number: CN2023106783413), which relates to the spectrometer design presented in this manuscript. This patent was also pursued as a pending PCT application (application number: PCT/CN2023/116853). The authors declare no other competing interests.
