## [Peer Review File · Nature Communications]

Integrated reconstructive spectrometer with programmable photonic circuitsReviewer #1 (Remarks to the Author):

- Congratulations to the authors on submitting a paper about redesigning a silicon nitride photonic spectroscopic IC. The authors have used the technique of reconstructive spectroscopy to claim a spectroscopic sensor that can provide a bandwidth-resolution ratio of 20,000. In short, the authors claim to break the bandwidth-resolution limits of wavelength de-multiplexing spectroscopic systems and Fourier transform spectrometers.
- The authors claim the device to be programmable. I fail to understand how the device is programmable. The authors use thermal phase shifters to achieve 729 sampling channels. For that, they use three different phase states. I hope they are not calling it "programmability". It is a mere adjustment or tuning of the device. I would highly recommend the authors clarify it, please. Generally, programmable photonic ICs provide different functionalities (for example, a filter whose architecture (FIR, IIR, ARMA, etc.) can be programmed to achieve different spectral responses and free-spectral ranges, etc.).
- The authors have used the term "limited on-chip resources" in the paper. What do they mean by that?
- The authors claim that the WDSs have inherently limited bandwidth. How come? Is it due to the cyclic nature of these spectroscopic systems? Can it please be elaborated on? If one designs an "AWG", where the delay length in the grating arms is not constant, would it also become broadband as the FSR will not repeat?
- The author may revisit the literature as the current manuscript does not mention all relevant published results.
- Line 112 has a typo. The authors may double-check the manuscript for other instances. In general, it is a well-written paper.
- Line 162 claims a full manipulation of the transmission spectrum. Can it be shown on a pole-zero diagram? Such as the position of the poles (real and imaginary parts) of the MZI can be moved to any position on a unit circle.
- What is the bandwidth of the directional couplers?
- How do you define the bandwidth of the spectrometer?
- Why do the authors limit themselves to using three-phase states? Use of four states does not change the footprint, system complexity, etc. Can it be shown experimentally how the performance of the RS varies by using 1, 2, 3, and 4 phase states? And does these experimental findings align with the simulation results?
- Apart from the change of coupling ratio for the directional couplers due to dispersion, the delay lines of the six MZI also suffer from a lack of dispersion engineering. Does it need to be discussed in the manuscript? For example, does the RS impact from the change of the group index for the delay lines of the MZI?
- Figure 3(d) needs to be improved.
- Figure 4 is difficult to read. Font size could be better.
- What is the efficiency of the thermal heaters? Is there any cross-talk between different stages? If not, can it be shown?
- Figure 5: does not include all relevant reported results. Further due diligence is necessary. Once again, I would like to congratulate the authors for their hard work and for drafting a good paper. I hope they will consider improving it further by revising it.

Reviewer #2 (Remarks to the Author):

This work focuses on development of a reconstructive spectrometer which is claimed to have a spectral resolution of a few picometers. A few cascaded MZI were used to temporally generate hundreds of spectral responses. This design is novel and the work is interesting. But a few comments should be addressed before it can be accepted.

1. This work defines the spectral resolution by resolving two closely spaced peaks. Such definition of spectral resolution deserves discussion. I know there are many relevant papers also define the spectral resolution in the same way. But in practical applications, the spectra to be reconstructed is seldom two peaks. Thus, it is more appropriate to reconstruct a spectrum with many peaks. Those peaks are separated with different spaces. Then, the resolution can be defined as the

minimum distance that can be resolved between all these peaks.

2. Also, when reconstructing two peaks to define the spectral resolution. If they can be resolved according to the Rayleigh criterion, but the peak values are far away from the correct number, does that mean the peaks are resolved. What is the role of the reconstruction accuracy when defining the spectral resolution?

3. The symbol C in equation 5 seems conflict with the symbol C in Line 140. Are they the same?

4. What is the exact value of the parameter C used to generate Fig. 1a. How is this value determined?

5. In Line 188, $\Delta\lambda$ is determined to be 0.58 nm, this is a few orders larger than the claimed spectral resolution which is a few picometers. Why $\Delta\lambda$ is so big, but you can still reconstruct peaks with such narrow width. Then what is the meaning of this parameter $\Delta\lambda$.

6. What is the overall light efficiency of the spectrometer. How does the efficiency changes as the number of stages increases.

7. I appreciate the motivation of Fig. 5a. But again, the definition of spectral resolution should be consistent across all these works. For example, the dispersion type of spectrometer does not require reconstruction, thus the spectrum does not suffer from errors such as artefacts. It is always reliable as long as measurement noise is low. But this is not true for reconstructive ones, even noise level is low, there are still inaccuracies caused by reconstruction error. Is it really meaningful to compare the spectral resolution of all those types? Maybe it is easier to compare between reconstructive ones?

8. Figure 5b is a bit confusing. Since both the x and y axis contains N_{spectral} , if we divide both axis with N_{spectral} , the figures would be the same? If so, then what is the role of N_{spectral} here? Does it play a role?

9. Line 292, please spell out SOI.

10. Please comment on the mass production of the spectrometer.

11. It is nice to briefly mention applications where picometer spectral resolution is required. for example, tunable diode laser spectroscopy can be used to realize such resolution. Much higher sensitivity can be achieved with lasers. What is the advantage of this method?

12. Figures are not very clear. High resolution figures should be used.

13. To ensure the reproducibility of this work, the relevant code and data should be uploaded for readers' interest. For example, the data to produce figure 4 should be uploaded. The numerical file for the design should be uploaded as well.

Reviewer #3 (Remarks to the Author):

This manuscript describes a novel concept of integrated spectrometer based on cascaded programmable interferometers and spectral reconstruction. The demonstrated devices feature a combination of resolution and bandwidth which goes well beyond the state of the art and their performance is convincingly proved in a number of experiments. Some (not all) of the limitations are also properly analyzed. It may be considered for publication if the following shortcomings are addressed:

- The title Redesigning spectroscopic sensors is unclear and confusing. "Spectroscopic sensor" is not a standard term in the literature, and the proposed device is not a sensor (it does not sense anything) but rather a spectrometer. I suggest using the keywords compressive spectrometer and programmable circuits and also avoiding "redesigning" which is vague and does not bring useful information

- The introduction and conclusion should be fully revised in the part referring to applications. The proposed device is based on single-mode waveguides, the coupling efficiency is negligible when using spatially-incoherent light from diffuse reflectance. This makes it completely useless for applications such as healthcare monitoring, smartphone-based spectral sensing or even biomedical sensing of glucose as claimed by the authors. The authors should be transparent with regards to this limitation and mention only applications which involve a single-mode input, such as the interrogation of fiber Bragg gratings or channel monitoring in telecom systems

- The sketch of Fig. 2a is not clear. In the region between the MZIs a connection is shown only for the upper waveguides, but presumably also the lower waveguides are connected. Also the position of the directional couplers should be more clearly labeled. The arrows should be more visible and

their direction checked, it is not clear why in the lower arm the arrows point in opposite directions. Please also make sure that the SEM of Fig, 3a has sufficient resolution in the final version so that details can be appreciated

- In the text the authors refer to "tunable resonant cavities" but it is not clear what these cavities are. In the structure light seem to flow from left to right without forming a folded path.
- In equation (8) a proper scientific notation should be used

Dear Reviewers,

Thank you so much for your valuable comments. We feel that the resulting amendments to our manuscript have substantially increased the quality of it. Below, please find our responses to the queries of the reviewer 1,2 and 3. Note that we use blue text for our responses to the reviewers, red text when quoting text from the manuscript, and highlighted red text to identify new or modified text in the manuscript. We also attach the revised manuscript with highlights that identify the changes.

Reviewer #1 (Remarks to the Author):

Congratulations to the authors on submitting a paper about redesigning a silicon nitride photonic spectroscopic IC. The authors have used the technique of reconstructive spectroscopy to claim a spectroscopic sensor that can provide a bandwidth-resolution ratio of 20,000. In short, the authors claim to break the bandwidth-resolution limits of wavelength de-multiplexing spectroscopic systems and Fourier transform spectrometers.

We sincerely appreciate the reviewer's detailed and constructive comments. We believe the manuscript is significantly improved by the revisions outlined below in response to the reviewers' comments.

1. The authors claim the device to be programmable. I fail to understand how the device is programmable. The authors use thermal phase shifters to achieve 729 sampling channels. For that, they use three different phase states. I hope they are not calling it "programmability". It is a mere adjustment or tuning of the device. I would highly recommend the authors clarify it, please. Generally, programmable photonic ICs provide different functionalities (for example, a filter whose architecture (FIR, IIR, ARMA, etc.) can be programmed to achieve different spectral responses and free-spectral ranges, etc.).

Many thanks for the valuable suggestions, and we apologize for any confusion in the original manuscript. By the term "programmability", we meant beyond the setting of phase states for individual MZI stages, but the capability of our device to adapt its performance metrics, such as the resolution, reconstruction precision or computation time, to suit the varying requirements of different application scenarios. Specifically, as illustrated in Fig. 1 and Fig. 4(f), the global sampling strategy allows the incident signals to be reconstructed using different sampling channels targeting at different levels of resolution and accuracy. Since our design can be fully controlled to generate different numbers of high-performance, decorrelated sampling channels, the users can easily customize the device to achieve performance trade-offs, which is a unique advantage compared with conventional reconstructive spectrometers with passive sampling strategies.

However, we agree that for the proposed spectrometer, different performance metrics cannot be fully manipulated as they are inevitably tied with each other. Therefore, we believe that "versatility" would be a more accurate term to describe such a feature. Accordingly, we have now made relevant updates across the paper. c

Introduction:

The reconfigurability of our design offers users the versatility to customize the device to achieve performance trade-offs on resolution, reconstruction accuracy, sampling time, and computational complexity by grouping different combinations of sampling channels, covering the application scenarios from identifying signature spectral peaks with acceptable levels of performance³⁰ to relative metrology with ultra-high resolution and low errors³¹.

Results:

As previously discussed, the global sampling strategy of RS allows a fewer number of sampling channels to coarsely reconstruct the whole input spectrum. This feature naturally offers our device versatility that trade-offs could be made between various performance metrics, such as the reconstruction accuracy and computational complexity. To investigate their underlying relationship, we analyze the reconstruction error and computing time as a function to the channel number, using the ASE spectrum of EDFA as an example case.

Discussion

Leveraging the global sampling scheme, our design features versatility that trade-offs can be made among the reconstruction performance, sampling time, and computational complexity by grouping different combinations of sampling channels, resulting in a user-definable performance to suit different applications.

2. The authors have used the term “limited on-chip resources” in the paper. What do they mean by that?

We appreciate the inquiry. In our original Introduction, the term "limited on-chip resources" refers to the physical limitations tied to the design and fabrication of on-chip spectrometers, such as footprint, count of building blocks, and/or overall power consumption.

To better clarify this, we have now replaced the term “limited on-chip resources” in the Introduction, as:

With the advances in nano-fabrication technologies, these schemes were later applied to photonic integrated circuits (PICs) for on-chip spectrometry, enabling significantly reduced device footprints⁵. Nevertheless, due to physical constraints on available chip area, number of building blocks, and/or power consumption, their performance, especially the resolution and bandwidth, is inevitably bounded.

3. The authors claim that the WDSs have inherently limited bandwidth. How come? Is it due to the cyclic nature of these spectroscopic systems? Can it please be elaborated on? If one designs an “AWG”, where the delay length in the grating arms is not constant, would it also become broadband as the FSR will not repeat?

Again, thanks for the insightful questions. The reason that we claim the traditional WDSs suffer from inherent bandwidth-resolution limitation is they rely on a one-to-one linear mapping between the spectral pixels and the detection channels (either in spatial or temporal domain), i.e., the bandwidth-to-resolution ratio being equal to the channel number. Therefore, for a WDS to approach a large bandwidth-to-resolution ratio, it is necessitated to employ either a large scale of building blocks (such as a vast array of micro ring filters and/or detectors) or a fine-tuning system that requires delicatied control with a long sequential detection time. Both schemes are to demultiplex the overall incident spectrum and cut the power of incident signal into portions that equal the bandwidth-to-resolution ratio. This means that the resolution will be ultimately bounded by the minimum detectable power (due to the limited signal-to-noise ratio of the PDs), which sets the fundamental limitation. Such limitation also applies to an AWG even if it could have an unlimited FSR.

Hence, following the revisions we made in response to the previous comment, we further revised the relevant sentences in Introduction, which goes as:

With the advances in nano-fabrication technologies, these schemes were later applied to photonic integrated circuits (PICs) for on-chip spectrometry, enabling significantly reduced device footprints⁵. Nevertheless, due to physical constraints on available chip area, number of building blocks, and/or power consumption, their performance, especially the resolution and bandwidth, is inevitably bounded.

For example, the WDSs, including those based on arrayed waveguide gratings⁶, echelle diffraction gratings⁷, or tunable narrowband resonators⁸⁻¹⁰, spectrally decompose the incident light into spatial or temporal detection channels. This results in a strict one-to-one linear mapping between the spectral pixels and channel number. Consequently, the pixel number, i.e. the bandwidth-to-resolution ratio, is ultimately bounded by the minimum detectable power per channel.

4. The author may revisit the literature as the current manuscript does not mention all relevant published results.

We appreciate reviewer's suggestion and have now included more relevant publications, please kindly find the details regarding the further added references in our responses to the Comment 14.

5. Line 112 has a typo. The authors may double-check the manuscript for other instances. In general, it is a well-written paper.

We thank the reviewer for noting the typo at line 112. We have now corrected the typo and thoroughly reviewed the manuscript to avoid further instances, as:

The concept of compressive sensing (CS) ~~can~~ **can** be borrowed to help understand the reconstruction in underdetermined systems.

6. Line 162 claims a full manipulation of the transmission spectrum. Can it be shown on a pole-zero diagram? Such as the position of the poles (real and imaginary parts) of the MZI can be moved to any position on a unit circle.

Many thanks to the constructive suggestion. We acknowledge that the pole-zero diagram is a widely used tool to describe the full response of optical filters. However, we would like to point out that these diagrams are primarily used for coherent signals, where both amplitude and phase are of significant importance. In contrast, for spectrometers, the purpose is generally to decode the amplitude/intensity of incident signal over wavelength and the detection is incoherent. In our design, the "phase tuning" of the MZI stages refers to the spectral shifting of their waveforms, thereby de-correlating different sampling channels. As such, we achieve "full manipulation" of the sampling responses by both tailoring the free spectral range and extinction ratio of the MZIs and tuning their phase shift in the range from $-\pi$ to $+\pi$ to realize a low cross-correlation among channels.

Accordingly, we have made relevant revisions in the main text. Moreover, we have now included a new paragraph in the Supplementary Section 2 to discuss the phase tuning and other key thermal characteristics of the MZIs, including an extra figure that displays the measured phase shift under different driving powers. The corresponding revisions are as follows:

Main text (Page 4, paragraph 2):

Therefore, by engineering the spectral properties of each MZI, such as the free spectral range (FSR) and extinction ratio (ER), **and phase-tuning of individual MZI stage, the waveform of the** overlaid transmission spectrum can be fully manipulated. Specifically, we introduce distinct length differences in various MZI stages to create interferograms with different FSRs, such that the cascaded interferogram no longer displays the original periodicities and exhibits a pseudo-random fluctuation in the wavelength domain, as shown by the insets in Fig. 2(a). The ERs are also tailored to maximize the fluctuation range of the cascaded interferogram, facilitating a high sampling efficiency with low excess loss. Thermo-optic (TO) phase shifters are implemented on the MZI arms **to tune the phase of each interferogram, thereby de-correlating different sampling channels**. Thus, by temporally setting different combinations of phase shifts, the cascaded interferogram can exhibit unique spectra responses, yielding a transmission

matrix with low cross-correlation, as shown by Fig. 2(b). Further details regarding the MZIs' parameter design, TO phase tuning, and power efficiency can be found in Supplementary Section S2.

Supplementary Section 2:

To facilitate precise phase tuning of each MZI stage, we conduct calibration processes by launching a laser signal into the cascaded system and sweeping the driving power of each phase shifter individually, while monitoring the output optical power. As an example, the inset in Fig. S3 presents the measured optical power in relation to the driving power applied to the heater on the long arm of the first MZI stage. The smooth sinusoidal curve not only indicates the phase shift, but also confirms the absence of thermal crosstalk between stages. Accordingly, we determine the relationship between the phase shift and driving power across all 6 MZI stages, as shown in Fig. S3. For illustrative purposes, we refer to the driving power applied on the long arm of the unbalanced MZIs as positive, which correlates with a positive phase shift, and vice versa. It can be seen that the complementary phase shifter pair on both MZI arms realizes the phase modulation from $-\pi$ to $+\pi$, with a thermal efficiency of around 42 mW/rad. Hence, in our experiment, the average power is about 350 mW as the phase shifters are set to induce phase shifts of either $-2\pi/3$, 0, or $2\pi/3$. Such thermal efficiency can be notably enhanced by incorporating deep trenches or undercuts adjacent to the waveguide^{S7}. Meanwhile, the implementation of our design on a Si platform could also significantly elevate the thermal efficiency due to its over 10-times superior thermo-optic coefficient.

Figure S3 | Thermal phase tuning of the unbalanced MZIs. The phase shift as a function of the driving power across all 6 MZI stages. Note that here the driving power applied to the long arm of the unbalanced MZIs is denoted as positive, leading to a resultant positive phase shift, and vice versa. As an example, the inset shows the measured optical power as a function of the driving power applied to the phase shifter located on the long arm of the first MZI stage.

7. What is the bandwidth of the directional couplers?

Thanks for the question. In our design, the bandwidth of the directional coupler is closely related to the spectrometer system as a whole. Its splitting ratio, denoted as ρ , is used to determine the optimal extinction ratio of the MZIs, and is dependent on the number of stages. We carefully tailor the splitting ratio ρ to ensure that the overlaid channel spectral response achieves maximum amplitude fluctuation over the targeted bandwidth range. As an example, we simulate the spectral response of both a 6-stage and 8-stage designs with varying ρ , as shown by Fig. S2(a-b) in the Supplementary Section 2, as:

Figure S2 | Parameter design and simulation. (a-b) Examples of the simulated transmission spectra for a 6-stage and an 8-stage programmable spectrometer with different power splitting ratio ρ of the MZIs, respectively. (c) Schematic of a symmetric DC. (d) Simulated wavelength dependence of the transmittance of the optimized symmetric DC.

As shown by Fig S2(a), for a 6-stage design, a ρ of around 0.10 is mostly preferred, as it allows the superimposed waveform to best fluctuate between 0 and 1. Even if ρ deviates, its pseudo-random spectral fluctuations can still be well generated, though with a slightly compromised range. Accordingly, we engineer ρ to be around 0.1 by fine-tuning its geometrical parameters. The FDTD simulated results are depicted in Fig. S2(d), showing that ρ maintains at around 0.1 (between 0.07 to 0.13) over a 200 nm spectral range from 1410 nm to 1610 nm. Further simulation results show that the ρ remains between 0.05 to 0.15 across the spectral range from 1390 nm to 1650 nm. Hence, the functional bandwidth of our directional coupler is defined at least over 200 nm, where the effective spectral fluctuations are produced.

Please note that such bandwidth could be further improved by employing dispersion-engineered curved directional coupler (instead of the symmetrical direction coupler). As shown by Fig. S2(e-f), such curved directional coupler could achieve a consistent splitting ratio between 0.06 to 0.15 over a 400 nm wavelength range, as:

Figure S2 | Parameter design and simulation. (e) Schematic of a curved DC. (f) Simulated wavelength dependence of the transmittance of the optimized curved DC.

To make these points more explicit to the readers, we have now re-written the relevant contents in both the main text and supplementary material, as follows:

Main text (Page 6, paragraph 3):

Figure 3(d) plots several examples of the sampling channels, showing the pseudo-random fluctuations in the wavelength domain. Here, the calibration is performed within a 200 nm spectral window from 1410 nm to 1610 nm which is limited by the bandwidth of the ASE source - the actual bandwidth of the spectrometer is expected to be larger. Accordingly, all measured spectra are reconstructed over such a 200 nm spectral range. The minor overall downward trend in the transmission can be attributed to the dispersion effect of the directional coupler, which exhibits a slight increase in splitting ratio at longer wavelengths. This can be further improved by employing dispersion-engineered components, such as curved directional couplers, to facilitate a flat splitting ratio over a broader bandwidth range. Detailed discussion of various directional coupler designs can be found in Supplementary Section S2. Moreover, in Supplementary Section S3, we show that by employing curved directional couplers, the device bandwidth could be further extended to over 400 nm.

Supplementary Section 2:

Figure S2(d) presents the FDTD simulated power transmittance at various output ports as a function of the wavelength, revealing a stable splitting ratio ρ of around 0.1 that varies slightly from around 0.07 to 0.13 across a spectral range exceeding 200 nm. To further broaden such bandwidth, dispersion-engineered waveguide components can be utilized, such as the curved DC^{S6}, as shown by Figure S2(e). Figure S2(f) shows the simulated port transmittances across a 400 nm bandwidth from 1250 nm to 1650 nm. It can be seen that the splitting ratio ρ remains consistently around 0.1 (between 0.06 and 0.15).

8. How do you define the bandwidth of the spectrometer?

Again, we thank for this question concerning the device bandwidth. Following answers to the comment above, we have shown that our current direction coupler design can work up to over 260 nm wavelength range. In our experiments, due to the limited bandwidth of our ASE light source, we could only verify a 200 nm operating bandwidth range from 1410 nm to 1610 nm. This is the reason we claim the spectrometer bandwidth to be only 200 nm in the manuscript.

Accordingly, please find the relevant clarifications in the main text (Page 6, paragraph 3), as:

Figure 3(d) plots several examples of the sampling channels, showing the pseudo-random fluctuations in the wavelength domain. Here, the calibration is performed within a 200 nm spectral window from 1410 nm to 1610 nm which is limited by the bandwidth of the ASE source - the actual bandwidth of the

spectrometer is expected to be larger. Accordingly, all measured spectra are reconstructed over such a 200 nm spectral range.

9. Why do the authors limit themselves to using three-phase states? Use of four states does not change the footprint, system complexity, etc. Can it be shown experimentally how the performance of the RS varies by using 1, 2, 3, and 4 phase states? And does these experimental findings align with the simulation results?

Many thanks for the questions. Our design indeed allows an exponential growth in the number of sampling channels by the increase of phase states per MZI. However, setting 3 phase states per stage in our 6-stage device (yielding 729 channels) is thoughtfully decided to maintain the balance among the device performance, computational complexity, and measurement time.

First of all, we are not able to produce stable dual laser peaks with a spectral spacing at single-digit picometers in our lab, while 729 sampling channels have been proven to be sufficient to achieve a < 10 pm resolution. Instead, we have performed rigorous simulations to investigate the cases when employing more phase states, as detailed in Supplemental Section 2 and 3. The results show that by increasing the phase tuning state P_{state} of the first four stages from 3 to 4 (thereby generating $4^4 \times 3^2 = 2304$ channels), the spectrometer resolution can be improved to <5 pm. This aligns with the results shown in Fig 1(a) that the spectrometer's resolution can be progressively improved with the increase of sampling channels. A secondary reason is the increased sampling time and computation complexity, as shown by Fig. 4(f). For example, if we were to apply 4 phase states across all 6 MZI stages, it would result in 4096 channels with a large scale of transmission matrix exceeding 4096×50000 .

Accordingly, we have now made the following revisions in the main text, as (Page 6, paragraph 3; Page 9, paragraph 2):

Figure 3(a) shows the microscope photo of the fabricated spectrometer. In this demonstration, we implement a 6-stage design with 3 phase settings per stage (i.e. 729 sampling channels in total) as a balanced choice among the device performance, measurement time and computational complexity.

To investigate their underlying relationship, we analyze the reconstruction error and computing time as a function to the channel number, using the ASE spectrum of EDFA as an example case. Note that here the experiments are conducted by varying the number of phase states across different MZI stages to attain different number of channels. As illustrated in Fig. 4(f), the reconstruction error gradually decreases with a larger number of sampling channels, albeit with increasing computing time. Hence, the performance of our design could be user-defined to suit the requirements under different scenarios¹⁷.

10. Apart from the change of coupling ratio for the directional couplers due to dispersion, the delay lines of the six MZI also suffer from a lack of dispersion engineering. Does it need to be discussed in the manuscript? For example, does the RS impact from the change of the group index for the delay lines of the MZI?

We appreciate the valuable questions. As discussed in our responses to Comments 6-9, the channel sampling responses is characterised only by the amplitude of waveforms. The dispersion in the MZI delay lines induces phase variations over wavelength, thus has little influence on the device performance given the incoherent detection. Also, any effects from waveguide dispersion, would be incorporated as part of spectral responses that are translated into the sampling matrix of the device during calibration. Therefore, in our case, the dispersion effects in delay lines do not affect the spectrometer's performance.

Accordingly, we have included corresponsive discussion in the revised manuscript, as (Page 6, paragraph 3):

Note that the dispersion from the connecting waveguides has a negligible impact on the device performance, as it is incorporated as part of spectral responses.

11. Figure 3(d) needs to be improved.

We apologize that the previous version of Fig. 3(d) appeared somewhat distorted. We have now addressed this issue, and adjusted the font size and letter spacing for better clarity, as:

Figure 3 | Fabricated spectrometer and its calibration. (d) Transmission spectra of several representative sampling channels, exhibiting the pseudo-random fluctuations in the wavelength domain.

12. Figure 4 is difficult to read. Font size could be better.

We appreciate this observation and have uniformed the font size in Fig. 4, as:

Figure 4 | Experimental testing results of the spectrometer.

13. What is the efficiency of the thermal heaters? Is there any cross-talk between different stages? If not, can it be shown?

Thanks for the great questions. As mentioned in our response to Comment 6, our phase shifters on the SiN platform demonstrated a thermal efficiency of approximately 42 mW/rad. Hence, in our experiment, the average power is around 350 mW, as the phase shifters are driven to deliver either $-2\pi/3$, 0, or $2\pi/3$ phase shifts. The thermal efficiency can be easily enhanced by many established fabrication techniques, such as incorporating deep trenches or undercuts to the waveguide (Ref S7). These methods are routinely offered by various foundries. The transition of our design onto a Si platform could also largely

improve the thermal efficiency, given its over 10-times superior thermo-optic coefficient. We have now included the corresponding discussions in Supplementary Section 2, as:

Supplementary Section 2:

It can be seen that the complementary phase shifter pair on both MZI arms realizes the phase modulation from $-\pi$ to $+\pi$, with a thermal efficiency of around 42 mW/rad. Hence, in our experiment, the average power is about 350 mW as the phase shifters are set to induce phase shifts of either $-2\pi/3$, 0, or $2\pi/3$. Such thermal efficiency can be notably enhanced by incorporating deep trenches or undercuts adjacent to the waveguide^{S7}. Meanwhile, the implementation of our design on a Si platform could also significantly elevate the thermal efficiency due to its over 10-times superior thermo-optic coefficient.

Reference:

S7. De, S., Das, R., Varshney, R. K. & Schneider, T. Design and Simulation of Thermo-Optic Phase Shifters With Low Thermal Crosstalk for Dense Photonic Integration. *IEEE Access* 8, 141632–141640 (2020).

For the perspective of thermal crosstalk, in our device, the distance between two adjacent MZI is designed to be over 400 μm , which gives rise to a negligible level of crosstalk between different stages (also thanks to the thermal robustness of SiN). Moreover, no thermal crosstalk observed during the calibration of MZI stages experimentally - please see the measured smooth sinusoidal curve in the inset of Figure S3. Relevant discussions can be found in the updated Supplementary Section 2, as:

As an example, the inset in Fig. S3 presents the measured optical power in relation to the driving power applied to the heater on the long arm of the first MZI stage. The smooth sinusoidal curve not only indicates the phase shift, but also confirms the absence of thermal crosstalk between stages.

14. Figure 5: does not include all relevant reported results. Further due diligence is necessary. Once again, I would like to congratulate the authors for their hard work and for drafting a good paper. I hope they will consider improving it further by revising it.

Again, many thanks to the comments. We have revisited the pertinent literature and enriched Figure 5 with all relevant studies, especially those published after the submission of our manuscript. The updated figure 5 is as follows:

Figure 5 | Performance comparison with state-of-the-art competitors. Comparison with various reported spectrometers regarding (a) the resolution and bandwidth, (b) the spectral pixel-to-footprint ratio and the spectral pixel-to-spatial detection channel ratio respectively.

Reference:

16. Xu, H., Qin, Y., Hu, G. & Tsang, H. K. *Breaking the resolution-bandwidth limit of chip-scale spectrometry by harnessing a dispersion-engineered photonic molecule.* *Light Sci Appl* **12**, 64 (2023).
38. Zhang, Z. et al. *Compact High Resolution Speckle Spectrometer by Using Linear Coherent Integrated Network on Silicon Nitride Platform at 776 nm.* *Laser & Photonics Reviews* **15**, 2100039 (2021).
39. Yao, C. et al. *Broadband picometer-scale resolution on-chip spectrometer with reconfigurable photonics.* *Light Sci Appl* **12**, 156 (2023).
60. Yi, D., Zhang, Y., Wu, X. & Tsang, H. K. *Integrated Multimode Waveguide With Photonic Lantern for Speckle Spectroscopy.* *IEEE J. Quantum Electron.* **57**, 1–8 (2021).
61. Du, J. et al. *High-resolution on-chip Fourier transform spectrometer based on cascaded optical switches.* *Optics Letters* **47**, 218–221 (2022).
62. Lin, Z. et al. *High-performance, intelligent, on-chip speckle spectrometer using 2D silicon photonic disordered microring lattice.* *Optica* **10**, 497–504 (2023).
63. Piels, M. & Zibar, D. *Compact silicon multimode waveguide spectrometer with enhanced bandwidth.* *Scientific reports* **7**, 43454 (2017).

Reviewer #2 (Remarks to the Author):

This work focuses on development of a reconstructive spectrometer which is claimed to have a spectral resolution of a few picometers. A few cascaded MZI were used to temporally generate hundreds of spectral responses. This design is novel and the work is interesting. But a few comments should be addressed before it can be accepted.

We are sincerely grateful for the reviewer's valuable comments. Accordingly, we have undertaken careful revisions to enhance the robustness and quality of our study. Detailed responses are provided below.

1. This work defines the spectral resolution by resolving two closely spaced peaks. Such definition of spectral resolution deserves discussion. I know there are many relevant papers also define the spectral resolution in the same way. But in practical applications, the spectra to be reconstructed is seldom two peaks. Thus, it is more appropriate to reconstruct a spectrum with many peaks. Those peaks are separated with different spaces. Then, the resolution can be defined as the minimum distance that can be resolved between all these peaks.

Many thanks for the suggestion. As noted by the reviewer, determining the spectrometer's resolution via the reconstruction of two closely spaced peaks is a commonly used method in this field. However, we agree that validating the resolution through the inclusion of multiple peaks would be more convincing. Hence, we have conducted additional experiments to investigate such a scenario. Owing to limitations in our laboratory equipment, we have tested the case involving three laser peaks, where a spectral spacing of 10 pm exists between two of them. The reconstructed results maintain a low relative error ϵ of 0.060 and 0.074, respectively. The slightly higher reconstruction error can be attributed to the increased noise level collectively originating from the third laser source.

Accordingly, we have made relevant revisions in the new manuscript, as (Page 7, paragraph 2):

Beyond the dual-peak experiment, we conduct a more challenging triple-peak testing, wherein the spectral spacing among two peaks is set at 10 picometers. Figure 4(c) shows the resolved three laser peaks with a relative error of 0.074 and 0.060, respectively.

Figure 4 | Experimental testing results of the spectrometer. (c) Reconstructed spectra for three spectral lines at different wavelengths, with a spectral spacing of 10 pm between two of them.

2. Also, when reconstructing two peaks to define the spectral resolution. If they can be resolved according to the Rayleigh criterion, but the peak values are far away from the correct number, does that mean the peaks are resolved. What is the role of the reconstruction accuracy when defining the spectral resolution?

We thank again for the insightful question. The valid reconstruction of a dual-peak signal (i.e., the verification of Rayleigh criterion) involves not only resolving the spectral locations of the two peaks

but also accurately determining their intensities. Thus, to evaluate reconstruction accuracy, we employ the L2-norm aggregated relative error, ε , as defined in Eq. 6. This metric is widely recognized and utilized in the literature. Typically, a relative error ε lower than 0.1 indicates successful reconstruction, as supported by many references (e.g., please see Ref 10, 26, 37, 39, 41). In our paper, we strictly follow such a benchmark. For example, all our dual-peak experiments demonstrate relative errors lower than 0.056, illustrating that the peak intensities can be well distinguished. Another counterexample is shown by Fig. S3 in Supplementary Section 3, where the reconstruction of dual laser peaks with a 7 pm spectral spacing results in a relative error of 0.28, which, therefore, is categorized as a failed reconstruction. Furthermore, it should be noted that failed reconstructions of dual-peak signals often appear as a single peak with an excessively high power, or as multiple peaks with distorted values, which results in a significant increase in errors, clearly marking the reconstructions as unsuccessful.

To clarify this, we have made relevant revisions and updated the references (Page 4, paragraph 1), as:

Here, the L2-norm relative error is utilized to quantify the spectrum reconstruction accuracy as a widely adopted metric^{2,10}, which is defined as follows:

$$\varepsilon = \frac{\|\Phi_0 - \Phi\|_2}{\|\Phi_0\|_2} \quad (6)$$

where Φ is the reconstructed spectrum and Φ_0 is the reference. Notably, a relative error lower than 0.1 is typically considered as a benchmark indicative of high accuracy^{10,26,37,39,41}.

Reference:

37. Wan, Y., Fan, X. & He, Z. Review on Speckle-Based Spectrum Analyzer. *Photonic Sens* 11, 187–202 (2021).

39. Yao, C. et al. Broadband picometer-scale resolution on-chip spectrometer with reconfigurable photonics. *Light Sci Appl* 12, 156 (2023).

41. Kurokawa, U., Choi, B. I. & Chang, C.-C. Filter-Based Miniature Spectrometers: Spectrum Reconstruction Using Adaptive Regularization. *IEEE Sensors Journal* 11, 1556–1563 (2011).

3. The symbol C in equation 5 seems conflict with the symbol C in Line 140. Are they the same?

We are grateful for the reviewer's observation. Indeed, the symbols 'C' used in Equation 5 and Line 140 are not the same – the former signifies the auto-correlation function as $C(\Delta\lambda)$, while the latter denotes a constant. To ensure clarity, we have now replaced the 'C' in Line 140 with the symbol 'A' to denote the constant, whilst maintaining 'C' for the auto-correlation function throughout the manuscript. The relevant revisions are as follows (Page 3, paragraph 1; Page 4, paragraph 1):

As revealed by CS theories, the minimum number of sampling channels (i.e. M) required to reconstruct an N -dimension incident spectrum is proportional to $A \log N$, where A is a constant related to the design of transmission matrix^{34,35}

This finding agrees with the CS theory as $M \sim \mathcal{O}(A \log N)$. Specifically, when N is relatively small and A could be considered as a low-value constant, M exhibits a gradual but slow growth as N expands over long intervals, reflecting the logarithmic relationship between M and N . As N grows larger, the changes in A become more pronounced, such that even a small increment in N results in a substantial increase in M (see further discussions in Supplementary Section S1).

4. What is the exact value of the parameter C used to generate Fig. 1a. How is this value determined?

Many thanks for the valuable question. For the simulations in Fig. 1a, our aim was to investigate the impact of channel number on the resolution and reconstruction accuracy. Therefore, we randomly generate a group of transmission matrices with different channel numbers. The half-width-half-maximum of the auto-correlation function $C(\Delta\lambda)$ of these matrices (i.e., the $\delta\lambda$) is consistently set as 0.2 nm. This particular value was chosen only for investigative purposes, as it represents a feasible benchmark for on-chip reconstructive spectrometers, in line with many published studies (e.g., please see Ref. 18, 20, 26, 39). Detailed explanations about how we generated the transmission matrices with consistent $\delta\lambda$ of 0.2 nm is provided in the Supplemental information S1, as:

As discussed in the main body text, a high-performance transmission matrix should have a small auto-correlation width $\delta\lambda$ and low cross-correlation between channels. This indicates that for the most ideal transmission matrix, its auto-correlation function $C(\Delta\lambda)$ should be a Dirac-delta function as it features the smallest possible auto-correlation width $\delta\lambda$, i.e., $\lambda_{bandwidth}/N$, while the cross-correlation should approach zero. Mathematically, the elements of such matrix (i.e., with Dirac-delta auto-correlation function and zero cross-correlation) ought to be independent identically distributed (i.i.d.) random variables, which is, however, unattainable for any physical optical structures. Hence, to generate a quasi-ideal transmission matrix that is achievable, we first create an i.i.d. random matrix that follows $\lambda_{bandwidth}/N = 0.2$ nm, and then fit the discrete elements in each row (i.e., each sampling channel) using cubic spline interpolation to make them continuous. By this manner, transmission matrices with a consistent auto-correlation width $\delta\lambda$ of 0.2nm are obtained, as illustrated by Fig. S1(a-b). Utilizing these matrices, we explore the relationship between the sampling channel number M and the resolution/spectral pixel N , as depicted in Fig. 1(a).

To avoid any confusion to the readers, we have now made responsive revisions in the main text (Page 3, paragraph 2), which goes as follows:

Therefore, to explore the performance boundary of RSs as a guide for practical designs, we randomly generate a group of broadband, continuous sampling channels with rapid spectral fluctuations over a 200 nm wavelength range (hence featuring low $\delta\lambda$ and cross-correlation) to approximate the “ideal” transmission matrix, as shown in Fig. S1(a) in Supplementary Section S1. The $\delta\lambda$ of all matrices is set at 0.2 nm, while the averaged cross-correlation is below 0.05, serving as a reasonable benchmark for reconstructive spectrometers^{18,20,26}. Detailed procedures for generating these matrices are provided in Supplementary Section S1

5. In Line 188, delta_lambda is determined to be 0.58 nm, this is a few orders larger than the claimed spectral resolution which is a few picometers. Why delta_lambda is so big, but you can still reconstruct peaks with such narrow width. Then what is the meaning of this parameter delta_lambda.

Again, we thank for the valuable questions. The auto-correlation function $C(\Delta\lambda)$ represents the correlation length of channel sampling responses in terms of the wavelength lag. And the half-width-half-maximum (HWHM) of $C(\Delta\lambda)$, denoted as $\delta\lambda$, can be used to represent the wavelength lag that is sufficient to reduce the strength of correlation by 50%. Hence, the value of $\delta\lambda$ indicates how rapid the intensity of sampling responses fluctuates over wavelength, such that a smaller value of $\delta\lambda$ usually signifies a better capability of distinguishing adjacent wavelength pixels (i.e. a higher resolution).

However, as revealed by many other published papers (e.g. the Ref 37-39) as well as our study in this paper, it is evident that the actual reconstructed resolution is jointly determined by many other factors including number of sampling channels, algorithm performance, and OSNR, etc. Importantly, our study, from the fundamental math perspective, illustrates that the actual resolution can significantly exceed the value of $\delta\lambda$ when a sufficient number of sampling channels is employed. For example, as shown by Fig. 1(a), while the $\delta\lambda$ is consistently 0.2 nm, the resolved resolution is able to reach single-digit

picometers. This finding agrees with the compressive sensing theory as $M \sim \mathcal{O}(\text{Alog}N)$, i.e., the resolved spectral pixel number N is positively related to the channel number M . Therefore, the $\delta\lambda$ should be considered a performance indicator, but not a determinant of the spectrometer's resolution.

Moreover, the temporal sampling nature of our design also ensures its maximal sampling OSNR despite of the large channel numbers, significantly contributing to the ultra-high resolution. In contrast, RS designs based on passive splitting or filtering schemes struggle to improve their resolution by simply adding more channels, since doing so inevitably compromises the OSNR as the detectable light per channel degrades.

Accordingly, we have included further explanation in the revised manuscript to clarify this (page 3, paragraph 1), as:

Thus, $\delta\lambda$ is usually regarded as an important performance indicator for the sampling channels³⁶. However, it should be noted that the resolution of the spectrometer system is jointly determined by many other factors including number of sampling channels, algorithm performance, signal-to-noise ratio (SNR), and etc^{37–39}. In the following, we show that the resolved resolution can significantly exceed the value of $\delta\lambda$ by employing a sufficient number of sampling channels.

Reference:

37. Wan, Y., Fan, X. & He, Z. Review on Speckle-Based Spectrum Analyzer. *Photonic Sens* **11**, 187–202 (2021).

38. Zhang, Z. et al. Compact High Resolution Speckle Spectrometer by Using Linear Coherent Integrated Network on Silicon Nitride Platform at 776 nm. *Laser & Photonics Reviews* **15**, 2100039 (2021).

39. Yao, C. et al. Broadband picometer-scale resolution on-chip spectrometer with reconfigurable photonics. *Light Sci Appl* **12**, 156 (2023).

6. What is the overall light efficiency of the spectrometer. How does the efficiency changes as the number of stages increases.

We appreciate the helpful question. We would like to provide a loss analysis detailing the overall light efficiency of the device. The temporal sampling nature ensures our device does not suffer from passive splitting loss. The edge couplers are designed to have a mode diameter of about 3.5 μm to match the UHNA fiber array, resulting in a coupling loss of < 3.2 dB per facet. The on-chip loss of the 6-stage device is measured to be around 4.9 dB, denoting the loss per MZI is about 0.8 dB (attributed to its filtering loss, insertion loss, and propagation loss of connecting waveguide). Therefore, introducing an additional MZI stage is at a cost of extra 0.8 dB loss.

Relevant analysis has now been included in the Method section, which goes as follows:

Device fabrication, design, and loss analysis. The spectrometer was fabricated via a CORNERSTONE SiN multi-project wafer (with a 300 nm LPCVD SiN layer) run using standard DUV lithography (250 nm feature size). The waveguides are designed to support TE polarization with a width of 1200 nm. The edge couplers are designed to have a mode diameter about 3.5 μm to match the UHNA fiber array, resulting in a coupling loss of < 3.2 dB per facet. The measured on-chip loss of the 6-stage device is about 4.9 dB, corresponding to a loss of roughly 0.8 dB per stage.

7. I appreciate the motivation of Fig. 5a. But again, the definition of spectral resolution should be consistent across all these works. For example, the dispersion type of spectrometer does not require reconstruction, thus the spectrum does not suffer from errors such as artefacts. It is always reliable as long as measurement noise is low. But this is not true for reconstructive ones, even noise level is low,

there are still inaccuracies caused by reconstruction error. Is it really meaningful to compare the spectral resolution of all those types? Maybe it is easier to compare between reconstructive ones?

We value the reviewer's thoughtful comments and agree that the reconstruction process in reconstructive spectrometers may inherently introduce algorithm-related inaccuracies. First of all, as in line with our response to the reviewer's Comment 1 and 2, we regard the Rayleigh criterion as the only metric to verify the resolution of any type of spectrometer. Therefore, we only included the reconstructive spectrometers that have conducted strict dual-peak experiments (i.e., the Rayleigh criterion) with a high level of accuracy in the table for a fair comparison. Secondly, the purpose of Fig. 5 is to better summarize the demonstrated specifications of the state-of-the-art miniaturized spectrometer designs, aiming to help readers better understand the whole picture. To prevent any misinterpretation to the readers, we have now highlighted the differences between various spectrometer types, and emphasized that we only include those reconstructive spectrometers that have been validated in Fig. 5a, which goes as follows (Page 10, paragraph 1):

Figure 5(a) plots the resolution and bandwidth (as two most important metrics) of integrated spectrometers based on dispersive optics, narrowband filtering, Fourier transform and computational reconstruction, respectively, along with our device^{7,9,11,13,14,16,18,21,28,38,39,51-63}. Note that only reported spectrometers with claimed resolutions that are experimentally verified by dual-peak testing (i.e. the Rayleigh criterion) with high reconstruction accuracies are included. As can be seen, the previously reported spectrometers exhibit clear trade-offs between the bandwidth and resolution, resulting in bandwidth-to-resolution ratios that mostly range from tens to hundreds.

Figure 5 | Performance comparison with state-of-the-art competitors. Comparison with various reported spectrometers regarding (a) the resolution and bandwidth, (b) the spectral pixel-to-footprint ratio and the spectral pixel-to-spatial detection channel ratio respectively.

8. Figure 5b is a bit confusing. Since both the x and y axis contains N_{spectral} , if we divide both axis with N_{spectral} , the figures would be the same? If so, then what is the role of N_{spectral} here? Does it play a role?

Thanks for the questions and apologize for any confusion. In Fig. 5(b), the y axis represents the ratio between the number of spectral pixels (N_{spectral}) and device footprint, while the x axis represents the ratio between the number of spectral pixels (N_{spectral}) and the number of spatial detection channel. However, since the value of N_{spectral} is unique to each individual spectrometer, it is impossible to treat it

as a constant and divide both axes by it. We select these two insightful ratios for evaluation because they effectively illustrate the capacity of spectral pixels that can be accommodated per unit of footprint or per individual detector unit, respectively. To avoid any further confusion, we have made relevant revisions, as (Page 10, paragraph 1):

We also evaluate the spectrometer performance in terms of the ratio between the number of spectral pixel (denoted as $N_{spectral}$) to device footprint, and the ratio of the number of spectral pixels to the number of spatial detection channel (denoted as $N_{detection\ channel}$), as shown by Fig. 5(b). The two terms offer valuable insights showing the capacity of spectral pixels that can be accommodated per unit area or per individual detector, respectively¹⁶.

9. Line 292, please spell out SOI.

We appreciate the reviewer's recommendation for clarification and have spelled out the term "SOI", as (Page 10, paragraph 1):

By implementing the same design on a standard 220 nm silicon-on-insulator (SOI) platform⁴³, the footprint is anticipated to be reduced to $< 0.4\text{ mm}^2$, which would further enhance the spectral pixel-to-footprint ratio by an order of magnitude, as illustrated in Fig. 5(b).

10. Please comment on the mass production of the spectrometer.

Thank you for raising this important aspect. Our proposed PIC-based spectrometer is demonstrated on a silicon nitride (SiN) platform, which is fully compatible with the well-established Complementary Metal-Oxide-Semiconductor (CMOS) fabrication processes, allowing for large-scale, cost-effective production. Our design can also be readily implemented on other material platforms by adjusting the device structural parameters, such as the Si platform. Thereby, we have added the relevant elaborations in the Discussion section, as (Page 9, paragraph 5):

Demonstrated on a commercial SiN platform, our device is fully compatible with the developed Complementary Metal-Oxide-Semiconductor (CMOS) processes, allowing mass-production at low cost^{43,44}.

Reference:

43. Su, Y., Zhang, Y., Qiu, C., Guo, X. & Sun, L. Silicon Photonic Platform for Passive Waveguide Devices: Materials, Fabrication, and Applications. *Advanced Materials Technologies* **5**, 1901153 (2020).

44. Martens, D. et al. A low-cost integrated biosensing platform based on SiN nanophotonics for biomarker detection in urine. *Analytical methods* **10**, 3066–3073 (2018).

11. It is nice to briefly mention applications where picometer spectral resolution is required. for example, tunable diode laser spectroscopy can be used to realize such resolution. Much higher sensitivity can be achieved with lasers. What is the advantage of this method?

We thank reviewer's comment and have now further elaborated the possible application scenarios that require ultra-high spectral resolution in the Discussion section, as (Page 9, paragraph 5):

The ultra-high resolution of the proposed device opens doors to numerous applications with critical demands on precision, including the detection of methanol, water pollutants, or gas concentrations which may require a resolution of $< 0.1\text{ nm}$ or even 1 pm ^{2,45–48}, and the identification of subtle peak shifts for fiber Bragg grating sensors in monitoring temperature, strain, or chemical variations^{49,50}.

Reference:

45. Li, M., Xue, J., Du, Y., Zhang, T. & Li, H. Data fusion of Raman and near-infrared spectroscopies for the rapid quantitative analysis of methanol content in methanol–gasoline. *Energy & Fuels* **33**, 12286–12294 (2019).

46. Cabernard, L., Roscher, L., Lorenz, C., Gerdt, G. & Primpke, S. Comparison of Raman and Fourier Transform Infrared Spectroscopy for the Quantification of Microplastics in the Aquatic Environment. *Environ. Sci. Technol.* **52**, 13279–13288 (2018).

47. Hodgkinson, J. & Tatam, R. P. Optical gas sensing: a review. *Measurement science and technology* **24**, 012004 (2012).

48. Zhang, D., Wang, J., Wang, Y. & Dai, X. A fast response temperature sensor based on fiber Bragg grating. *Measurement Science and Technology* **25**, 075105 (2014).

49. Campanella, C. E., Cuccovillo, A., Campanella, C., Yurt, A. & Passaro, V. M. Fibre Bragg grating based strain sensors: Review of technology and applications. *Sensors* **18**, 3115 (2018).

50. Xu, B., Huang, J., Xu, X., Zhou, A. & Ding, L. Ultrasensitive NO Gas Sensor Based on the Graphene Oxide-Coated Long-Period Fiber Grating. *ACS Appl. Mater. Interfaces* **11**, 40868–40874 (2019).

On the other hand, tunable laser spectroscopy is indeed capable to achieve high spectral resolutions. However, our PIC-based spectrometer enjoys many unique advantages, including but not limited to the following:

- (1) Cost-Effectiveness: our design's compatibility with mature CMOS fabrication processes allows for mass production at much lower cost compared to tunable laser systems.
- (2) Broad Bandwidth: our device demonstrates a 200 nm bandwidth, which can be scaled further via dispersion engineering techniques to over 400 nm.
- (3) Control complexity: our device features a relaxed control scheme utilizing only few phase states per MZI, while the tunable laser spectroscopy systems necessitate precise wavelength tuning.

12. Figures are not very clear. High resolution figures should be used.

Many thanks to this feedback. We have now upgraded all the previous figures with high-resolution versions to improve their visibility and comprehension. In the latest submission, these high-resolution figures have also been submitted as distinct files for your reference.

13. To ensure the reproducibility of this work, the relevant code and data should be uploaded for readers' interest. For example, the data to produce figure 4 should be uploaded. The Lumerical file for the design should be uploaded as well.

We appreciate this suggestion and are fully supportive to ensure the reproducibility of our work. Hence, upon reasonable request from the readers, we're more than willing to share the necessary data, code, and the Lumerical simulation files regarding our spectrometer design.

Reviewer #3 (Remarks to the Author):

This manuscript describes a novel concept of integrated spectrometer based on cascaded programmable interferometers and spectral reconstruction. The demonstrated devices feature a combination of resolution and bandwidth which goes well beyond the state of the art and their performance is convincingly proved in a number of experiments. Some (not all) of the limitations are also properly analyzed. It may be considered for publication if the following shortcomings are addressed:

We thank the reviewer for the constructive comments. We have carefully addressed the indicated shortcomings and believe the manuscript is significantly strengthened through the revisions conducted following the reviewers' comments, which are detailed below.

1. The title Redesigning spectroscopic sensors is unclear and confusing. "Spectroscopic sensor" is not a standard term in the literature, and the proposed device is not a sensor (it does not sense anything) but rather a spectrometer. I suggest using the keywords compressive spectrometer and programmable circuits and also avoiding "redesigning" which is vague and does not bring useful information.

Thanks for the valuable suggestions regarding the title of our paper. In response, we have changed the title to "**Integrated reconstructive spectrometer with programmable photonic circuits**", which reflects the content and novel aspects of our study more accurately.

2. The introduction and conclusion should be fully revised in the part referring to applications. The proposed device is based on single-mode waveguides, the coupling efficiency is negligible when using spatially-incoherent light from diffuse reflectance. This makes it completely useless for applications such as healthcare monitoring, smartphone-based spectral sensing or even biomedical sensing of glucose as claimed by the authors. The authors should be transparent with regards to this limitation and mention only applications which involve a single-mode input, such as the interrogation of fiber Bragg gratings or channel monitoring in telecom systems.

We appreciate the reviewer's insightful comments. Indeed, coupling spatially incoherent light from diffuse reflectance into waveguides presents a significant challenge. This may impede certain applications, such as drone or satellite-based remote sensing or astronomical spectroscopy, where the light sources are inevitable diffuse and weak, so that spectrometers with large numerical aperture are needed to capture sufficient light.

However, in many applications – including the in-situ healthcare monitoring, chemical sensing, and even some optical imaging systems, like the Optical Coherence Tomography (OCT) – there is certain flexibility to have suitable broadband light sources with high power density and small mode diameters, such as the ASE sources or superluminescent diodes (SLDs), which can be readily coupled with single-mode fiber. Hence, it is feasible to pair such light source with our proposed PIC-based spectrometer to circumvent the challenges in light coupling. Specifically, the broadband light source, like SLDs, would be used to illuminate the samples (such as biological tissues, air or water samples, etc.), while the reflected/transmitted light can subsequently be collected by focusing lens and single mode fibers, and eventually coupled into the spectrometer chip. Such configuration has been reported by many papers, such as:

1). Rank, Elisabet A., et al. "Toward optical coherence tomography on a chip: in vivo three-dimensional human retinal imaging using photonic integrated circuit-based arrayed waveguide gratings." *Light: Science & Applications* 10.1 (2021): 6.

In this paper, the authors propose a PIC-based OCT system and demonstrate the in-vivo imaging of human retina. The system's setup is akin to the one described above, where a fiber-coupled SLD is used

for luminance and light reflected from the retina is then coupled into the SiN PICs with a ~ 2.5 dB fiber-to-chip coupling loss.

2). Martens, Daan, et al. "A low-cost integrated biosensing platform based on SiN nanophotonics for biomarker detection in urine." *Analytical methods* 10.25 (2018): 3066-3073.

Similarly, this paper presents a SiN photonic integrated sensor using MZIs and on-chip AWG spectral filters, where a SLD is used to illuminate urine samples, and the resulting transmitted light is collected by an on-chip grating coupler for spectral analysis.

Accordingly, we have carefully revised the Introduction and Discussion sections. Here, we list the key revisions:

Introduction

The past decades have witnessed an ever-growing demand for in situ, in vitro and in vivo spectroscopic measurement techniques to facilitate various applications, ranging from wearable healthcare monitoring, portable chemical sensing tools, to compact optical imaging systems^{1,2}. This trend has propelled the rapid progression of miniaturized optical spectrometers in both academia and industry.

Discussion

In summary, we believe the proposed design offers a promising pathway for the future development of low-cost, mass-manufacturable, chip-scale spectroscopic devices. Its ultra-high performance makes it hold great promise to play a role in a vast range of application areas.

3. The sketch of Fig. 2a is not clear. In the region between the MZIs a connection is shown only for the upper waveguides, but presumably also the lower waveguides are connected. Also the position of the directional couplers should be more clearly labeled. The arrows should be more visible and their direction checked, it is not clear why in the lower arm the arrows point in opposite directions. Please also make sure that the SEM of Fig. 3a has sufficient resolution in the final version so that details can be appreciated.

Many thanks to the detailed suggestions on Figure 2a and 3a. In our design, the MZI employed at each stage features a singular input and output, i.e., only one input/output port of the directional couplers is used. This configuration allows each MZI to operate individually as a periodic interferometer (as described by Eq. 7) and facilitates the pseudo-random spectral responses after cascading. Accordingly, we have revised the schematic to highlight this feature of our MZI design.

As for the arrows in Fig 2.a, we have also made them clearer to better reflect the actual light propagation. The revised Fig. 2a and corresponding amendments in the main text are provided below (Page 4, paragraph 2), as:

To efficiently meet such a requirement, here, we propose a programmable design cascading multiple stage of tunable interferometers. Figure 2(a) schematically shows the implementation of our design via a series of 1×1 unbalanced MZIs.

Figure 2 | Spectrometer design and simulation. (a) Schematic of our proposed spectrometer design with multiple stages of unbalanced MZIs, each with a singular input and output port. The insets show the increasing spectral randomness in the cascaded interferogram.

Meanwhile, we have upgraded the images in Fig. 3a with higher resolution, as can also be found in the re-submitted figure files:

Figure 3 | Fabricated spectrometer and its calibration. (a) Microscope image of the fabricated on-chip spectrometer in a 6-stage design with enlarged views of a MZI stage and a directional

4. In the text the authors refer to "tunable resonant cavities" but it is not clear what these cavities are. In the structure light seem to flow from left to right without forming a folded path.

We appreciate reviewer's attention to detail. We have revised the terminology throughout the manuscript by replacing the "resonant cavities" with "interferometers".

5. In equation (8) a proper scientific notation should be used

Many thanks for the suggestion, we have revised equation (8) to adopt proper scientific notation, which goes as:

As the phase shift of each MZI stage can be tuned individually from 0 to 2π , the overall tuning state (i.e., the total number of temporal sampling channels) features an exponential scalability, as:

$$N_{ch} = P_{state}^{N_{stage}} \quad (8)$$

Reviewer #2 (Remarks to the Author):

I appreciate the efforts of the authors. My comments have been addressed properly.

However, the authors recently (online on 25 June 2023) published a paper on Light Science and Applications
<https://www.nature.com/articles/s41377-023-01195-2>

Seems this work is very similar to the published one. The authors included this publication in the reference list i.e. Ref. 39. But they did not discuss the difference between the published work and this work under review.

To my understanding, the only difference is the spectral range, the resolution and a minor structural change to the MZIs.

The authors should clarify the novelty of this work. Currently, it seems a minor incremental compared with the one they published in Light Science and Applications. This of course do not deserve a second publication on a top journal.

Reviewer #3 (Remarks to the Author):

The authors have adequately addressed the points raised by me and the other reviewers, and the paper is suitable for publication in its current form.

Dear Reviewers,

Thank you again for the second-round review. Please find our responses to the latest comments. Note that we use *blue text* for our responses to the reviewers, *red text* when quoting text from the manuscript, and *highlighted red text* to identify new or modified text in the manuscript. We also attach the revised manuscript with highlights that identify the changes.

Reviewer #2 (Remarks to the Author):

I appreciate the efforts of the authors. My comments have been addressed properly.

However, the authors recently (online on 25 June 2023) published a paper on *Light Science and Applications* - <https://www.nature.com/articles/s41377-023-01195-> Seems this work is very similar to the published one. The authors included this publication in the reference list i.e. Ref. 39. But they did not discuss the difference between the published work and this work under review. To my understanding, the only difference is the spectral range, the resolution and a minor structural change to the MZIs. The authors should clarify the novelty of this work. Currently, it seems a minor incremental compared with the one they published in *Light Science and Applications*. This of course do not deserve a second publication on a top journal.

We thank the reviewer for the comment and we would like to point out that the design in the present work and the one in the recent *Light* paper are fundamentally different, ranging from the device structure, operational principle, to the performance and scalability.

In the *Light* paper (please see the following device schematic¹), the proposed spectrometer scheme relies on the distributed all-pass micro-ring resonators (MRRs) to facilitate distinct channel responses for spectral sampling. In this scheme, the meshes of balanced MZIs only operate as cross/bar switches, routing the incoming signal via different optical paths. Each path comprises a unique combination of MRR filters to yield diverse sampling responses. Thus, the MRR filters are the elements in creating the sampling spectral responses, while the MZI switches make no contribution to the responses. As a result, the number of sampling channels is tied to the scale of the mesh of switches, leading to a large footprint. Also, the MRR filters operate statically (i.e. with no thermal tuning), and the switching of the MZIs simply requires a π phase shift.

By contrast, in our current design, we utilize only unbalanced MZI filters with various oscillation periods to form the sampling responses. By setting each MZI in different phase states (e.g., $-2/3\pi$, 0 and $2/3\pi$ as demonstrated in our experiments), the overall sampling channel can be exponentially scaled up

Schematic of the demonstrated reconfigurable spectrometer based on meshes of MZI switches and distributed all-pass MRR filters¹.

¹ Chunhui Yao, et al. "Broadband picometer-scale resolution on-chip spectrometer with reconfigurable photonics." *Light: Science & Applications* 12.1 (2023): 156.

without any need to scale the device. This scheme remarkably improves from the one reported in *Light* based on passive MRRs and balanced MZI switches. Additionally, the current spectrometer features a much broader bandwidth, as the unbalanced MZI filters don't necessitate a strict 50:50 splitting ratio.

For better clarity, we have now included corresponding discussions in the Introduction part, which goes as:

Researchers have also explored active RSs using lumped structures with tunable sampling spectral responses, including detector-only RSs with tunable absorption spectra^{23,24}, filter-based RSs with MEMS²⁵ and thermally tunable resonance cavities^{26,27}. However, as the sampling channels are generated by setting different levels of driving powers to alter the waveform, they inevitably suffer from a poor decorrelation with each other, resulting in a compromised resolution (typically on the order of nanometers^{27,28}) and limited scalability. In our earlier work, we introduce a RS design that utilizes a reconfigurable network with distributed micro-ring resonator filters to create well-decorrelated sampling responses, while its bandwidth is constrained by the switching network, and the resolution has a trade-off against device complexity²⁹. So far, the reported on-chip RS schemes haven't shown a definitive performance advantage over the WDSs and FTSs.